# Recasting Transformer Layers as Energy Models

## Abstract

Foundation models rely on sequence-to-sequence mappings parameterized by neural networks, and the design space of these layers continues to expand. Transformer layers remain the dominant choice due to their strong performance and high parallelism, though many design decisions are still empirically based. We introduce Causal Energy Minimization (CEM), a framework that interprets each transformer layer as an algorithm for solving an energy minimization problem with causal structure. This perspective separates the mathematical interpretation of a layer from its numerical realization, offering a unifying lens for layer design and motivating principled architectural innovations. Within CEM, multi-head attention emerges as a gradient step on an interaction energy under the weight sharing constraint, while gated multilayer perceptrons (MLPs) correspond to element-wise energies. The form of transformer components within CEM suggests a weight-sharing scheme in both attention and MLP blocks: we show that this yields parameter-efficient layers with negligible performance loss. Further, the CEM interpretation suggests appealing extensions to the transformer architecture: preconditioner matrices for residual connections, diagonal matrices for inter-token distances in attention, and multiple gradient-steps (a form of layer reuse) for both attention and MLP blocks. We show that these ideas that occur naturally in CEM lead to improvements on language modelling tasks, positioning CEM as a blueprint for principled and extensible architecture design.

## 1 Introduction

Sequence-to-sequence mappings underlie modern foundation models (Bommasani et al., 2021). Early work employed recurrent (Sutskever et al., 2014; Hochreiter & Schmidhuber, 1997; Cho et al., 2014) and convolutional architectures (Kalchbrenner et al., 2016; Gehring et al., 2017), but these have been largely replaced by Transformer (Vaswani et al., 2017). While alternatives such as structured state-space models have recently emerged (Gu et al., 2022; Gu & Dao, 2023), Transformer layers, particularly multi-head attentions (MHAs) and gated MLPs, remain the core of today's large language models (LLMs). Yet architectural innovations for transformer continues to be driven mainly by empirical performance and efficiency (Shazeer et al., 2017; Shazeer, 2020; 2019; Ainslie et al., 2023). What is missing is a principled framework that explains why current choices work and provides systematic guidance for future innovation.

Energy-based models (EBMs) provide such a framework. By assigning a scalar energy $\mathcal{E}(\boldsymbol{x})$ to each configuration $\boldsymbol{x}$, EBMs define computation as the search for low-energy states (LeCun et al., 2006; Hopfield, 1982; Ackley et al., 1985; Krotov & Hopfield, 2016; Ramsauer et al., 2021). This formulation brings two key advantages. First, it offers explicit mathematical objectives: a model's computation is understood as minimizing a well-defined energy function, rather than applying a black-box transformation. Second, it connects architecture design to the theory of optimization, enabling analysis of stability and convergence. Yet, despite these advantages, standalone EBMs, especially in their classical and associative-memory forms, typically underperform Transformer on large-scale sequence modelling (Du et al., 2021; Qin & Eisner, 2022).

We introduce CEM, a framework that formulates each transformer layer as solving an energy minimization problem with causal structure. CEM separates the semantics of a layer (the energy it defines), from its numerical realization (the optimization algorithm used to minimize it). Transformer

layers such as MHAs and gated MLPs arise as special cases under weight-tying constraints: MHA corresponds to gradient steps on interaction energies, while gated MLP corresponds to element-wise energies (see Sections 2.1 and 2.2).

Concretely, to map an input sequence $\boldsymbol{h}_{1:J}$ to an output sequence $\boldsymbol{h}'_{1:J}$, CEM defines each output $\boldsymbol{h}'_i$ by introducing a variable $\boldsymbol{x}_i$, initialized at $\boldsymbol{h}_i$, and updating it with an optimization procedure $\mathcal{A}$ to (approximately) minimize a conditional energy $\epsilon(\boldsymbol{x}_i \mid \boldsymbol{h}_{1:i})$, which depends on the causal history $\boldsymbol{h}_{1:i}$. The optimized solution then becomes the new hidden state $\boldsymbol{h}'_i$. Here, the energy function interprets sequence processing, while the optimization algorithm specifies the numerical realization. Stacking such layers yields expressive sequence-to-sequence models.

**Contributions.** In this work, we study the Transformer architecture through the lens of CEM and make the following contributions:

- We show that MHA with weight sharing can be derived as a gradient step on an interaction energy, while gated MLP corresponds to an element-wise energy. This view allows weight sharing between up/down projections in MLPs and between linear projections in attention to arise naturally, leading to more parameter-efficient designs.
- Building on the energy optimization perspective, we extend transformer layer design beyond single gradient updates. We investigate diagonal-plus-low-rank weight matrices, preconditioned updates, multiple recursive steps.
- We show that CEM layers match the performance of larger Llama components while retaining interpretability through the energy-minimization framework. Moreover, optimization-driven design yields performance gains without increasing model size.

## 2 TRANSFORMER LAYERS AS ENERGY UPDATES

We start by reframing Transformer layers through the lens of CEM. We introduce two complementary energy terms: an *interaction term*, which captures dependencies across features for different tokens, and an *element-wise term*, which assigns energy to each token's feature vector. Taking gradient-based updates on these energies naturally recovers standard Transformer layers with weight sharing: the interaction updates yield multi-head attention, while the element-wise updates yield gated MLPs. Figure 1 presents an illustration of the weight sharing scheme.

### 2.1 GRADIENT OF INTERACTION ENERGY YIELDS WEIGHT-TIED ATTENTION

**Multi-head attention (MHA).** In conventional MHA, the query, key, and value projections for head $k$ are defined as

$$\boldsymbol{q}_i^k = \boldsymbol{W}_k^Q \boldsymbol{h}_i, \quad \boldsymbol{k}_j^k = \boldsymbol{W}_k^K \boldsymbol{h}_j, \quad \boldsymbol{v}_j^k = \boldsymbol{W}_k^V \boldsymbol{h}_j,$$

where $\boldsymbol{h}_j$ denotes $j$-th token feature vector in the sequence. The attention update then takes the form

$$\mathrm{MHA}(\boldsymbol{h}_{1:i}) = \sum_{k=1}^{K} \boldsymbol{W}_k^{O\top} \left( \sum_{j=1}^{i} \mathrm{softmax}_j \left( \left\{ \tfrac{1}{\sqrt{D_h}} (\boldsymbol{k}_{j'}^k)^\top \boldsymbol{q}_i^k \right\}_{j'=1}^{i} \right) \boldsymbol{v}_j^k \right).$$

Typically, the per-head outputs are concatenated and followed by a single output projection $\boldsymbol{W}^O$. Equivalently, one may view $\boldsymbol{W}^O$ as partitioned into head-specific blocks $\{\boldsymbol{W}_k^O \in \mathbb{R}^{D_r \times D_h}\}_{k=1}^{K}$, with contributions summed as written above, where $D_h$ is the feature dimension for $\boldsymbol{h}_i$ and $D_r$ is the head dimension where $\boldsymbol{q}_i^k, \boldsymbol{k}_j^k, \boldsymbol{v}_j^k \in \mathbb{R}^{D_r}$. See *Appendix A.1* for detailed explanation.

**Interaction energy.** MHA can be derived by considering a gradient step on the following simple interaction energy, similar to that in modern Hopfield networks (Ramsauer et al., 2021):

$$\epsilon(\boldsymbol{x}_i \mid \boldsymbol{h}_{1:i}) = -\tau \sum_{k=1}^{K} \log \sum_{j=1}^{i} \exp\left( \tfrac{1}{\tau} \boldsymbol{\beta}_{kj}^\top \boldsymbol{x}_i \right) \qquad \text{where} \qquad \boldsymbol{\beta}_{kj} = \boldsymbol{A}_k \boldsymbol{h}_j. \qquad (1)$$

Here $\{\boldsymbol{A}_k \in \mathbb{R}^{D_h \times D_h}\}_{k=1}^{K}$ are learnable projection matrices, $D_h$ is the feature dimension, and $\tau$ is a scalar temperature. Our formulation differs from Hopfield networks in two respects: projection

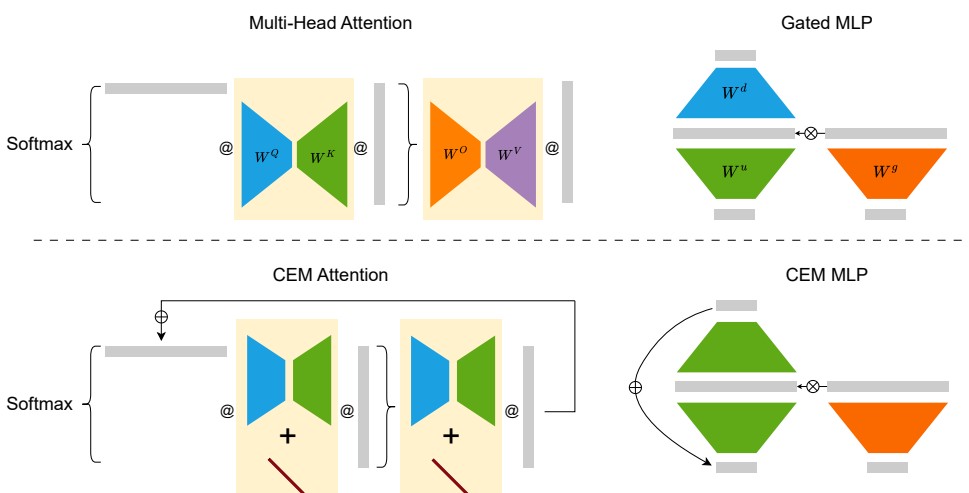

Figure 1: **Comparison of transformer layer parameterisations**. Top left: standard multi-head attention (per head). Top right: gated MLP. Bottom left: CEM-derived attention. Bottom right: CEM-derived MLP. Colors indicate shared weights within each subfigures (See Sections 2.1 and 2.2). Arrows highlight the recursive structure of CEM modules, which implement multiple gradient steps of energy minimization (Equations (16) and (17)), while brown bars denote the optional diagonal term added to the key–query projections (See details in Section 2.3). For the attention heads, $W^V$ maps the hidden state to values, which are then projected back by $W^O$ and scaled by the scalar Softmax weight.

weights are embedded directly in the energy and reappear as tied attention projections, and we perform gradient updates rather than Concave-Convex Procedure (CCCP) iterations (see Appendix C). We now derive the gradient of the interaction energy $\epsilon(\boldsymbol{x}_i \mid \boldsymbol{h}_{1:i})$ with respect to $\boldsymbol{x}_i$:

$$\nabla_{\boldsymbol{x}_i} \epsilon(\boldsymbol{x}_i \mid \boldsymbol{h}_{1:i}) = -\sum_{k=1}^{K}\sum_{j=1}^{i} \text{softmax}_j \left( \left\{ \tfrac{1}{\tau} \boldsymbol{\beta}_{kj'}^{\top} \boldsymbol{x}_i \right\}_{j'=1}^{i} \right) \boldsymbol{\beta}_{kj}. \tag{2}$$

Adopting a low-rank factorization $\boldsymbol{A}_k = \boldsymbol{W}_k^{Q\top} \boldsymbol{W}_k^{K}$ we obtain $\boldsymbol{\beta}_{kj} = \boldsymbol{W}_k^{Q\top}(\boldsymbol{W}_k^{K}\boldsymbol{h}_j)$. If our chosen algorithm is to take a single gradient step, initialized at $\boldsymbol{x}_i = \boldsymbol{h}_i$, then we compute:

$$\boldsymbol{h}_i' = \boldsymbol{h}_i - \eta_\epsilon \left. \nabla_{\boldsymbol{x}_i} \epsilon(\boldsymbol{x}_i \mid \boldsymbol{h}_{1:i}) \right|_{\boldsymbol{x}_i = \boldsymbol{h}_i}, \tag{3}$$

with

$$\left. \nabla_{\boldsymbol{x}_i} \epsilon(\boldsymbol{x}_i \mid \boldsymbol{h}_{1:i}) \right|_{\boldsymbol{x}_i = \boldsymbol{h}_i} = -\sum_{k=1}^{K} \boldsymbol{W}_k^{Q\top} \left( \sum_{j=1}^{i} \text{softmax}_j \left( \left\{ \tfrac{1}{\tau} (\boldsymbol{k}_{j'}^k)^{\top} \boldsymbol{q}_i^k \right\}_{j'=1}^{i} \right) \boldsymbol{v}_j^k \right). \tag{4}$$

where the value and key projections are shared: $\boldsymbol{q}_i^k = \boldsymbol{W}_k^{Q}\boldsymbol{h}_i$, $\boldsymbol{v}_j^k = \boldsymbol{k}_j^k = \boldsymbol{W}_k^{K}\boldsymbol{h}_j$.

It is therefore clear that the gradient of the interaction energy recovers the MHA form, with the weight-tied parameterization

$$\boldsymbol{W}_k^{K} = \boldsymbol{W}_k^{V} \qquad \boldsymbol{W}_k^{Q} = \boldsymbol{W}_k^{O} \qquad \tau = \sqrt{D_h}. \tag{5}$$

In this view, the residual update corresponds exactly to a single gradient descent step (with step size $\eta^\epsilon = 1$) on the defined interaction energy.

## 2.2 GRADIENT OF ELEMENT-WISE ENERGY YIELDS WEIGHT-TIED MLPS

**Gated MLPs.** A gated MLP applies an element-wise transformation to the hidden state $\boldsymbol{h}_i \in \mathbb{R}^{D_h}$:

$$\text{GatedMLP}(\boldsymbol{h}_i) = \boldsymbol{W}^{\text{d}}\big((\boldsymbol{W}^{g}\boldsymbol{h}_i) \circ \sigma(\boldsymbol{W}^{u}\boldsymbol{h}_i)\big). \tag{6}$$

Here, the learnable parameters are the *gate* and *up* projections $\boldsymbol{W}^{g}, \boldsymbol{W}^{u} \in \mathbb{R}^{D_m \times D_h}$ and the *down* projection $\boldsymbol{W}^{\text{d}} \in \mathbb{R}^{D_h \times D_m}$. The function $\sigma$ denotes a pointwise nonlinearity (e.g. GELU).

**Element-wise energy term.** This energy term assigns energy independently to each token feature vector, while sharing the same functional form across positions:

$$\xi(\boldsymbol{x}_i \mid \boldsymbol{h}_i) = -\boldsymbol{\gamma}_i^\top \phi(\boldsymbol{V}\boldsymbol{x}_i), \qquad \text{where} \qquad \boldsymbol{\gamma}_i = \boldsymbol{W}\boldsymbol{h}_i. \tag{7}$$

Here, the learnable parameters are the projection matrices $\boldsymbol{W}, \boldsymbol{V} \in \mathbb{R}^{D_v \times D_h}$, with projection dimension $D_v$ not necessarily equal to the hidden dimension $D_h$. The function $\phi$ denotes a pointwise nonlinearity.

**Energy-gradient formulation.** For the element-wise energy $\xi$, the gradient with respect to $\boldsymbol{x}_i$ is

$$\nabla_{\boldsymbol{x}_i} \xi(\boldsymbol{x}_i \mid \boldsymbol{h}_i) = -\boldsymbol{V}^\top \big(\boldsymbol{\gamma}_i \circ \phi'(\boldsymbol{V}\boldsymbol{x}_i)\big). \tag{8}$$

Taking one gradient step at $\boldsymbol{x}_i = \boldsymbol{h}_i$ yields

$$\boldsymbol{h}_i' = \boldsymbol{h}_i - \eta_\xi \, \nabla_{\boldsymbol{x}_i} \, \xi(\boldsymbol{x}_i \mid \boldsymbol{h}_i)\Big|_{\boldsymbol{x}_i = \boldsymbol{h}_i}, \tag{9}$$

with

$$\nabla_{\boldsymbol{x}_i} \xi(\boldsymbol{x}_i \mid \boldsymbol{h}_i)\Big|_{\boldsymbol{x}_i = \boldsymbol{h}_i} = -\boldsymbol{V}^\top \big((\boldsymbol{W}\boldsymbol{h}_i) \circ \phi'(\boldsymbol{V}\boldsymbol{h}_i)\big). \tag{10}$$

Comparing equation 6 and equation 10, the energy-gradient update recovers the structure of gated MLPs when we identify the parameters as

$$\boldsymbol{W}^{d\top} = \boldsymbol{W}^u = \boldsymbol{V}, \qquad \boldsymbol{W}^g = \boldsymbol{W}, \tag{11}$$

and we set $\phi(x) = \int_{-\infty}^x \sigma(z)\mathrm{d}z$.

## 2.3 Enhancing transformer layers from an energy optimization perspective

Having shown that Transformer layers, both MHA and MLPs, can be interpreted as gradient updates on energy functions in Sections 2.1 and 2.2, we next explore how these layers can be enhanced from the perspective of energy optimization.

**Diagonal-plus-low-rank parameterisation** . In Section 2.1, we introduced a low-rank parameterisation of $\boldsymbol{A}_k = \boldsymbol{W}_k^{Q\top} \boldsymbol{W}_k^K$ in the interaction energy, recovering the query and key projections of standard attention. We now ask whether a purely low-rank form is sufficient, and instead propose a diagonal-plus-low-rank parameterisation for the matrix $\boldsymbol{A}_k$:

$$\boldsymbol{A}_k = \mathrm{diag}(\boldsymbol{d}_k) + \boldsymbol{W}_k^{Q\top} \boldsymbol{W}_k^K, \tag{12}$$

where $\boldsymbol{d}_k \in \mathbb{R}^{D_h}$. This augmented parameterisation captures key–query interactions that low-rank matrices alone cannot represent, yielding a richer structure for the interaction matrix $\boldsymbol{A}_k$. The diagonal term enriches the interaction matrix but increases computational cost, so we propose sharing it across heads. A detailed empirical analysis is provided in Figure 3b and more background on this parameterisation can be found in Appendix A.2.

**Learned lightweight preconditioners.** A single gradient descent step is often insufficient to reach a low-energy state. Second-order methods such as Newton's method accelerate convergence by scaling updates with the inverse Hessian, but computing and inverting Hessians is often expensive. We therefore introduce learned lightweight preconditioners: trainable low-rank positive-definite matrices that approximate curvature information in diagonal-plus-low-rank form,

$$\boldsymbol{P} = \mathrm{diag}\big(\mathrm{softplus}(\boldsymbol{d})\big) + \boldsymbol{U}\boldsymbol{V}^\top + \boldsymbol{V}\boldsymbol{U}^\top, \tag{13}$$

with $\boldsymbol{d} \in \mathbb{R}^{D_h}$ and $\boldsymbol{U}, \boldsymbol{V} \in \mathbb{R}^{D_h \times R}$, where $R \ll D_h$. Here $\mathrm{softplus}(x) = \log(1 + e^x)$ is applied element-wise to ensure strictly positive diagonal entries, guaranteeing $\boldsymbol{P} \succ 0$. For the interaction energy, we insert per-head preconditioners $\boldsymbol{P}_k$, giving the update (contrast with the unpreconditioned gradient in Equation (2)):

$$\Delta \boldsymbol{x}_i^\epsilon(\boldsymbol{h}_i \mid \boldsymbol{h}_{1:i}) := -\sum_{k=1}^K \boldsymbol{P}_k \, \boldsymbol{W}_k^{Q\top} \left( \sum_{j=1}^i \mathrm{softmax}_j \Big( \big\{ \tfrac{1}{\tau} (\boldsymbol{k}_{j'}^k)^\top \boldsymbol{q}_i^k \big\}_{j'=1}^i \Big) \boldsymbol{v}_j^k \right). \tag{14}$$

which denotes the update at $\boldsymbol{x}_i = \boldsymbol{h}_i$ for the energy $\epsilon$. For the element-wise energy, the gated MLP update becomes (contrast with unpreconditioned one in Equation (10)):

$$\Delta \boldsymbol{x}_i^\xi(\boldsymbol{h}_i \mid \boldsymbol{h}_{1:i}) \coloneqq -\boldsymbol{P}_{\text{mlp}} \boldsymbol{V}^\top \left( (\boldsymbol{W}\boldsymbol{h}_i) \circ \phi'(\boldsymbol{V}\boldsymbol{h}_i) \right), \tag{15}$$

with $\boldsymbol{P}_{\text{mlp}}$ denoting its preconditioner.

In both cases, the preconditioners could be trained to provide lightweight curvature information, enabling updates that converge more effectively to well-optimized states.

**Multiple recursive steps**  So far, each Transformer layer has been interpreted as performing a single gradient step on its associated energy function. From the optimization viewpoint, however, a single step rarely reaches a well-optimized state. A natural extension is therefore to apply multiple recursive updates within the same layer, analogous to running several iterations of an optimization algorithm. For the interaction energy (attention), starting from $\boldsymbol{x}_i^{(0)} = \boldsymbol{h}_i$, we perform $T$ updates of the form

$$\boldsymbol{x}_i^{(t+1)} = \boldsymbol{x}_i^{(t)} - \eta_\epsilon \, \Delta \boldsymbol{x}_i^\epsilon(\boldsymbol{x}_i \mid \boldsymbol{h}_{1:i}), \qquad t = 0, \ldots, T-1, \tag{16}$$

and set $\boldsymbol{h}_i' = \boldsymbol{x}_i^{(T)}$.

For the element-wise energy (MLP), starting from $\boldsymbol{x}_i^{(0)} = \boldsymbol{h}_i$, the recursion is

$$\boldsymbol{x}_i^{(t+1)} = \boldsymbol{x}_i^{(t)} - \eta_\xi \, \Delta \boldsymbol{x}_i^\xi(\boldsymbol{x}_i \mid \boldsymbol{h}_{1:i}), \qquad t = 0, \ldots, T-1, \tag{17}$$

with the output $\boldsymbol{h}_i' = \boldsymbol{x}_i^{(T)}$. This recursive scheme enables each layer to better minimize its energy function without adding parameters, as illustrated in Figure 1. Unlike blockwise recursion in looped Transformer, our approach updates only $\boldsymbol{x}_i^{(t)}$ and fix $\boldsymbol{h}_{1:i}$, with most computation performed outside the recursion. This within-layer recursion thus offers a distinct mechanism that could provide a new dimension for test-time scaling, which we leave for future work.

## 2.4  A CONSTRUCTION OF TRANSFORMER BLOCK WITH ENERGY UPDATES

We now present the full Transformer block from the CEM perspective, where both attention and MLP components arise as recursive gradient updates on their respective energy functions. Residual connections are absorbed into the recursion, while $\text{RMSNorm}(\cdot)$ are applied. Standard Transformer with weight sharing, as detailed in Equations (5) and (11), appears as the special case $T_\epsilon = T_\xi = 1$, using identity preconditioners $(\boldsymbol{P}_k) = (\boldsymbol{P}_{\text{mlp}}) = \boldsymbol{I}$ and vanishing diagonal terms $\boldsymbol{d}_k = \boldsymbol{0}$. The complete CEM block is summarized in Algorithm 1.

---

**Algorithm 1:** Transformer Block as Energy Updates (Orange parts highlight CEM specifics)

**Input:** Sequence $\boldsymbol{h}_{1:J}$,  **Output:** Sequence $\boldsymbol{h}_{1:J}'$
**Hyperparameters:** Recursive steps $T_\epsilon, T_\xi$, step sizes $\eta_\epsilon, \eta_\xi$, number of heads $K$,
$\phi(x) = \int_{-\infty}^x \text{SiLU}(z) \, \mathrm{d}z$
**Trainable parameters:** $\{\boldsymbol{W}_k^Q, \boldsymbol{W}_k^K, \boldsymbol{D}_k = \text{diag}(\boldsymbol{d}_k)\}_{k=1}^K, \boldsymbol{W}, \boldsymbol{V}, \{\boldsymbol{P}_k\}_{k=1}^K, \boldsymbol{P}_{\text{mlp}}$

| **Main Block** | **Subroutine: MHA** | **Subroutine: MLP** |
|---|---|---|
| **for** $i = 1 : J$ **do** | $\text{MHA}(\boldsymbol{h}_{1:i}, \boldsymbol{k}_{1:i}^k, \boldsymbol{v}_{1:i}^k)$: | $\text{MLP}(\boldsymbol{h}_i)$: |
| $\quad \boldsymbol{h}_{1:i} \leftarrow \text{RMSNorm}(\boldsymbol{h}_{1:i})$ | $\boldsymbol{x}_i \leftarrow \boldsymbol{h}_i$ | $\boldsymbol{x}_i \leftarrow \boldsymbol{h}_i$ |
| $\quad$ **for** $k = 1 : K$ **do** | **for** $t = 0 : T_\epsilon - 1$ **do** | $\gamma = \boldsymbol{W}\boldsymbol{h}_i$ |
| $\quad\quad \boldsymbol{k}_{1:i}^k \leftarrow \boldsymbol{W}_k^K \boldsymbol{h}_{1:i}$ | $\quad \boldsymbol{u}_i \leftarrow \text{RMSNorm}(\boldsymbol{x}_i)$ | **for** $t = 0 : T_\xi - 1$ **do** |
| $\quad\quad \boldsymbol{v}_{1:i}^k \leftarrow \boldsymbol{W}_k^K \boldsymbol{h}_{1:i}$ | $\quad$ **for** $k = 1 : K$ **do** | $\quad \boldsymbol{u}_i \leftarrow \text{RMSNorm}(\boldsymbol{x}_i)$ |
| $\quad \boldsymbol{h}_i \leftarrow$ | $\quad\quad \boldsymbol{q}_i^k \leftarrow \boldsymbol{W}_k^Q \boldsymbol{u}_i$ | $\quad \boldsymbol{g}_i \leftarrow \boldsymbol{V}^\top(\gamma \circ \phi'(\boldsymbol{V}\boldsymbol{u}_i))$ |
| $\quad\quad \text{MHA}(\boldsymbol{h}_{1:i}^{1:K}, \boldsymbol{k}_{1:i}^{1:K}, \boldsymbol{v}_{1:i}^{1:K})$ | $\quad\quad \boldsymbol{a}_{ijk} \leftarrow D_h^{-1/2}(\boldsymbol{k}_j^{k\top}\boldsymbol{q}_i^k + \boldsymbol{h}_j^\top \boldsymbol{D}_k \boldsymbol{u}_i)$ | $\quad \boldsymbol{x}_i \leftarrow \boldsymbol{x}_i + \eta_\xi \boldsymbol{P}_{\text{mlp}} \boldsymbol{g}_i$ |
| **for** $i = 1 : J$ **do** | $\quad\quad \boldsymbol{o}_i^k \leftarrow \sum_{j=1:i} \text{sftmx}_j(\{\boldsymbol{a}_{ijk}\}_{j=1}^i) \boldsymbol{v}_j^k$ | **return** $\boldsymbol{x}_i$ |
| $\quad \boldsymbol{h}_i \leftarrow \text{RMSNorm}(\boldsymbol{h}_i)$ | | |
| $\quad \boldsymbol{h}_i' \leftarrow \text{MLP}(\boldsymbol{h}_i)$ | $\quad \boldsymbol{x}_i \leftarrow \boldsymbol{x}_i + \eta_\epsilon \sum_k \boldsymbol{P}_k \boldsymbol{W}_k^{Q\top} \boldsymbol{o}_i^k$ | |
| **return** $\boldsymbol{h}_{1:J}'$ | **return** $\boldsymbol{x}_i$ | |

---

A subtlety arises when incorporating positional encodings: rotary embeddings (RoPE) in particular complicate the energy-gradient view by making the projection weights depend on both query and key indices. To avoid this overhead, we instead adopt relative-position biases such as Alibi, as discussed in Appendix B.

## 3  RELATED WORK

EBMs assign low energies to preferred configurations (Hopfield, 1982; LeCun et al., 2006). Modern extensions to Hopfield networks (Krotov & Hopfield, 2016; Ramsauer et al., 2021) with continuous patterns and log-sum-exp energy demonstrate how attention-like updates can arise from their updating iterations (Appendix C). Recent work on EBMs for language extend the formulation to sequence-level objectives, including residual EBMs for text generation (Du et al., 2021; Grathwohl et al., 2021), controllable decoding (Qin & Eisner, 2022; Liu et al., 2022). Our work reframes Transformer layers, including both MHA and MLP, as energy updates, and demonstrate that this perspective leads to principled extensions and improvements for text modeling tasks.

**Alternative transformer blocks.**   Transformer models have largely converged toward Llama-style backbones with multi-head attention (Vaswani et al., 2017) and gated MLPs (Shazeer, 2020). A wide range of efficiency-oriented variants aim to reduce the memory and compute cost of attention, for example through multi-query, group-query, or latent attention mechanisms (Shazeer, 2019; Ainslie et al., 2023; Zhai & et al., 2023). On the feedforward side, Liu et al. (2021); Shazeer (2020); So et al. (2021) demonstrate stronger and consistent empirical performance. Shazeer et al. (2017); Fedus et al. (2022) scale up capacity through sparsely activated mixture-of-experts (MoE) layers, He & Hofmann (2024); He et al. (2023) seek to streamline the architecture by simplifying skip connections, projections, or normalization layers with little or no degradation in performance.

**Recursive depth and adaptive computation.**   Layer sharing improves parameter efficiency, as demonstrated by the Universal Transformer (Dehghani et al., 2019), ALBERT (Lan et al., 2020), and Perceiver (Jaegle et al., 2021). Adaptive schemes further allocate computation dynamically through early exits or token-level routing (Elbayad et al., 2020; Xin et al., 2020; Bae et al., 2025), balancing efficiency and accuracy. Recursion also connects to latent reasoning such as latent chains of thought (Hao et al., 2024; Zhang & Viteri, 2024; Tan et al., 2025). In this work, we explore within-layer recursion, which offers an additional axis for adaptivity and can be combined with prior approaches.

## 4  EXPERIMENTS

Our experiments address four questions: (i) can CEM MHAs and MLPs act as parameter-efficient drop-in replacements for their standard counterparts; (ii) do within-layer recursion and lightweight preconditioners improve performance; (iii) can a Transformer composed entirely of CEM layers be trained end-to-end; and (iv) how do design choices such as KQ diagonal terms and recursion affect performance. All models are trained on SlimPajama for the compute-optimal number of tokens of the corresponding Llama baselines (Hoffmann et al., 2022), and we report test perplexity as the main evaluation metric. Experimental details can be found in Appendix D.

### 4.1  REPLACE TRANSFORMER LAYERS WITH SINGLE-STEP CEM LAYERS

To evaluate the effectiveness of CEM layers, we train Transformer models with CEM components in either the MLP or attention blocks, and compare against Llama baselines. We focus on the weight-tying formulation (see Equations 5 and 11), but without recursions or preconditioners here.

Figure 2a compares CEM attention with standard Llama MHAs, while Figure 2b compares CEM MLPs with Llama-style gated MLPs. Blue dots denote dimension-matched CEM models, where CEM attention uses about half the parameters and CEM MLPs about two-thirds of their Llama counterparts. Some degradation is expected, but the goal is to assess how closely CEM models approach baseline performance with fewer parameters. For CEM MLPs, we also report results with increased intermediate dimension to restore the baseline parameter count (orange triangles).

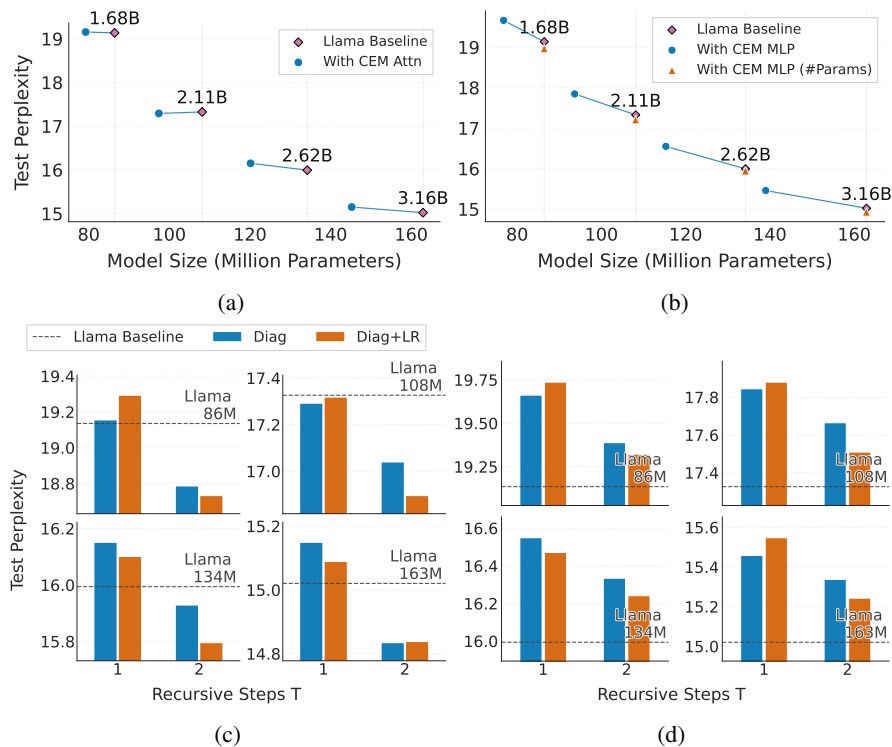

Figure 2: (a) Llama Transformer with attention replaced by CEM attention ($T = 1$). CEM variants (blue dots) are linked to their Llama baselines (pink diamonds) with matching dimensions but fewer parameters, trained on the same number of tokens (shown above markers). (b) Llama Transformer with gated MLPs replaced by CEM MLPs ($T = 1$). Orange triangles additionally show parameter-matched variants obtained by increasing the hidden dimension. (c) Effects of recursion steps (x-axis) and preconditioners (colors) for CEM attention, with all models dimension-matched to Llama baselines (dashed lines). (d) Effects of recursion steps and preconditioners for CEM MLPs.

Replacing attention with the CEM variant has only a small effect on test perplexity despite halving the parameter count, with no natural parameter-matching scheme available since the model dimension must remain fixed for controlled comparison. For CEM-MLPs, perplexity is higher due to parameter sharing, but increasing the hidden dimension to match parameter count yields consistent, albeit modest, improvements in perplexity — though at the cost of additional FLOPs. Unless otherwise noted, we adopt the optimal Llama hyperparameters from grid search (see Appendix D) to ensure consistent comparisons and avoid tuning each CEM configuration individually, even though these settings may be suboptimal for CEM models (see Figure 3a). These results indicate that single-step CEM layers can act as parameter-efficient drop-in replacements for standard Transformer components, achieving competitive perplexity with substantially fewer parameters, with CEM attention in particular showing more promising results that merit further investigation.

## 4.2 COMPARE RECURSIVE STEPS AND PRECONDITIONERS IN CEM LAYERS

We test whether within-layer recursion and lightweight preconditioners improve performance. As before, Transformer variants are trained on SlimPajama to the compute-optimal point of their Llama baseline and evaluated by test perplexity. We only study recursive steps until $T = 2$ because $T \geq 3$ has unstable performance and leads to OOM for larger model sizes we tested.

Figure 2c replaces standard Llama MHAs with CEM attention, while Figure 2d fixes the MHAs and instead replaces Llama MLPs with CEM MLPs. For each case, we compare diagonal and diagonal-plus-low-rank preconditioners, and evaluate $T = 1$ vs. $T = 2$. Dashed lines mark the Llama baselines. Across model sizes, we match dimensions, so CEM components always use fewer parameters; preconditioners add only a negligible overhead.

Results in Figure 2 show consistent gains when increasing recursion from $T = 1$ to $T = 2$. Preconditioners have little effect at $T = 1$ but yield clear improvements at $T = 2$. For attention layers, CEM attention with recursion and preconditioners not only remains more parameter-efficient but also significantly outperforms the Llama baseline. For MLPs, CEM variants still underperform the baseline, but the gap narrows considerably with $T = 2$ and a diagonal-plus-low-rank preconditioner. Overall, these results demonstrate that within-layer recursion ($T = 2$) reliably improves performance, while diagonal-plus-low-rank preconditioners provide additional, although modest, gains.

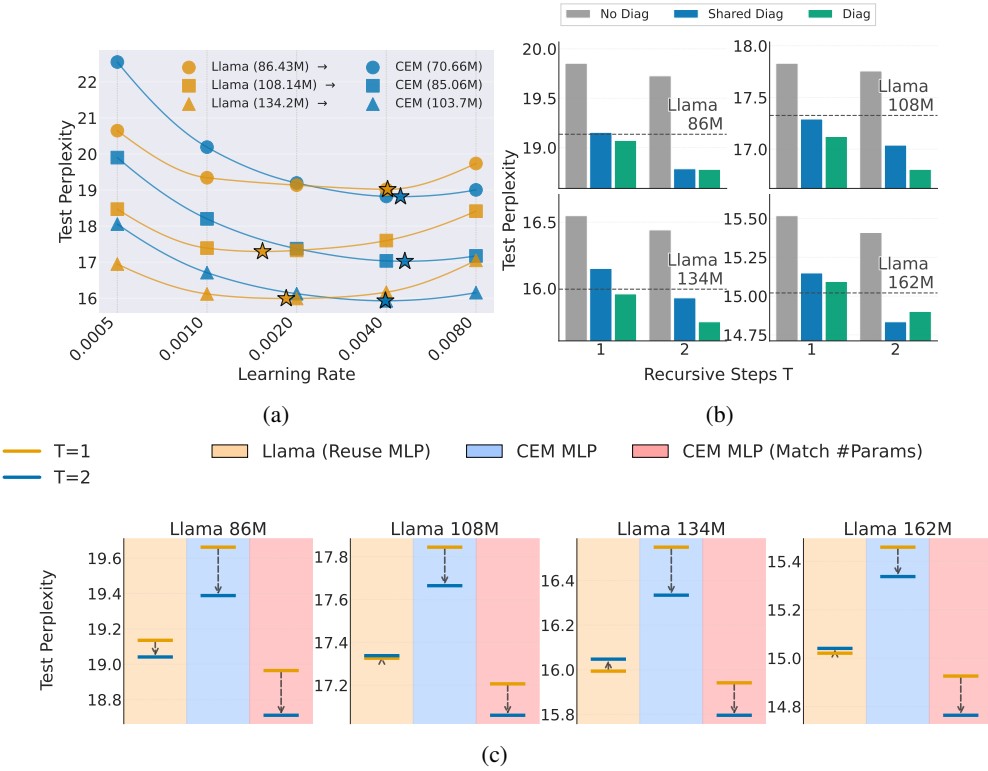

Figure 3: (a) Optimal learning rate estimated via Akima interpolation. Orange denotes baseline Llama models and blue denotes CEM models ($T = 2$) with both MHA and MLP replaced. Marker shapes indicate model size; stars mark interpolated optima from five data points. Matched Llama and CEM models (with roughly half the MHA and one-third the MLP parameters) are trained with the same token budget (Chinchilla-optimal for Llama). For smaller models, parameter reduction is less pronounced due to embedding and head parameters. (b) KQ diagonal strategies in CEM attention: no diagonal in $A_k$, a shared diagonal across heads, and per-head diagonals. All CEM models match the dimensionality of the Llama baselines (dashed line). (c) Within-layer recursion vs. plain layer reuse in MLPs. We compare the performance gains of increasing recursion from $T = 1$ (orange) to $T = 2$ (blue), under three settings: Plain layer reuse (light orange area), dimension-matched CEM MLP (blue area) and parameter-matched MLPs (pink area). An equivalent figure comparing within-layer recursion vs. layer reuse for MHA can be found in Figure 4.

## 4.3 TRAINING END-TO-END TRANSFORMER WITH CEM LAYERS

We have shown that CEM layers can serve as effective drop-in replacements for standard attention and gated MLPs. We now ask whether a Transformer built entirely from CEM modules can be trained end-to-end. Since $T = 2$ and diagonal-plus-low-rank preconditioners yielded the best performance for both CEM attention and MLPs, we adopt this configuration in the pure CEM-based transformer. In this setup, CEM attention uses about half the parameters of standard attention, and CEM MLPs about two-thirds, resulting in a more parameter-efficient architecture. Due to memory constraints, we omit the largest model with diagonal-plus-low-rank preconditioners.

We train models with five learning rates ranging from 0.0005 to 0.008, doubling at each step, and use Akima interpolation (Akima, 1970) to estimate the optimal rate. Figure 3a reports the interpolated optimal perplexity, where marker shapes denote model sizes and colors distinguish baseline Llama (orange) from CEM (blue). Actual model sizes are indicated in parentheses; for smaller models, the relative reduction is less pronounced due to embeddings and output heads.

Overall, full CEM Transformer achieves slightly better performance at their optimal learning rate while using considerably fewer parameters. Notably, CEM models tend to favor higher learning rates than their Llama counterparts.

### 4.4 Ablation study

We first study the role of diagonal terms in inter-token distances (Figure 3b) in attention, comparing three settings: no diagonal, a shared diagonal across heads, and per-head diagonals. All other components are fixed (Llama MLPs, CEM attention with one recursion step, and a simple diagonal preconditioner). Including a diagonal term proves essential for good performance, and our shared-diagonal strategy provides performance close to per-head diagonals while reducing parameters and compute, making it a more efficient alternative.

Second, we test whether within-layer recursion is necessary or if naive layer reuse suffices (Figure 3c). Simply reapplying the same residual block yields little or no perplexity gain. Note that this reuse differs from recursive Transformer, where entire blocks (attention and MLP) are reused. In contrast, CEM-based within-layer recursion produces consistent improvements in both dimension- and parameter-matched settings. A similar trend holds for attention (Figure 4).

## 5 Discussion and Conclusion

### 5.1 Limitations

Due to computational constraints, our experiments focus on models of around 100M parameters, leaving large-scale studies to future work. We also do not report downstream task performance, as such evaluations are most meaningful at larger scales. Nevertheless, test perplexity is a well-established proxy for downstream performance and provides a reliable measure of model quality.

While CEM layers are parameter-efficient, we have not yet explored custom kernels with weight sharing, fused operations, or hardware-specific optimizations that could further improve runtime efficiency. Reducing overhead, increasing throughput, and better aligning the design with modern accelerators remain important directions for future work.

### 5.2 Conclusion and future directions

We introduced CEM, a framework that recasts Transformer layers as energy-minimizing updates, yielding natural weight sharing, parameter-efficient architectures, and a principled path to new designs. Taking insights from this perspective, optimization-inspired enhancements including diagonal-plus-low-rank parameterizations, lightweight preconditioners, and within-layer recursion, can improve perplexity without increasing model size.

We think the following directions are worthwhile to explore further:

- **A new dimension for test-time scaling.** Explore whether CEM-style within-layer recursion can provide a new dimension for test-time compute scaling (Snell et al., 2024; Muennighoff et al., 2025) and latent reasoning (Hao et al., 2024; Zhang & Viteri, 2024; Tan et al., 2025), particularly when combined with blockwise recursion as in recursive or looped Transformer (Yang et al., 2023; Bae et al., 2024; Dehghani et al., 2019).
- **Custom kernels for CEM layers.** Develop FlashAttention-style kernels (Dao et al., 2022) for CEM layers by fusing tied projections, diagonal terms, and recursive updates into a single IO-aware kernel. Leveraging tiling, SRAM reuse, and fused epilogues can further reduce memory transfers and launch overhead, thereby improving both efficiency and throughput.
- **Architecture hardware co-design.** The CEM framework enables redesigning layers for unconventional hardware by rethinking the optimization procedure which yields novel layer

parameterization. For example, on photonic or analog accelerators, one can exploit native support for iterative solvers (Hua et al., 2025; Kalinin et al., 2025) to develop new building blocks that run natively and efficiently on such platforms.

## 5.3 REPRODUCIBILITY STATEMENT

We provide experimental details in Appendix D. Model architectures are given in Table 1, and training configurations in Table 2. All experiments use the SlimPajama dataset (Appendix D.2) and were conducted on 8× NVIDIA A100 GPUs. The code is not yet publicly available but will be released upon publication.

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

# A    BACKGROUND

## A.1    EQUIVALENCE BETWEEN CONCATENATION AND SUMMATION IN ATTENTION

In Section 2.1, we write the multi-head attention update in the form

$$\text{MHA}(\boldsymbol{h}_{1:i}) = \sum_{k=1}^{K} \boldsymbol{W}_k^{O\top} \left( \sum_{j=1}^{i} \text{softmax}_j \left( \left\{ \tfrac{1}{\sqrt{D_h}} (\boldsymbol{k}_{j'}^k)^\top \boldsymbol{q}_i^k \right\}_{j'=1}^{i} \right) \boldsymbol{v}_j^k \right),$$

where each head $k$ contributes an output vector that is multiplied by a head-specific block $\boldsymbol{W}_k^O \in \mathbb{R}^{D_r \times D_h}$.

This notation differs slightly from the conventional implementation of multi-head attention, where per-head outputs are concatenated and processed by a single output projection. To make the equivalence explicit, let

$$O_k := \sum_{j=1}^{i} \text{softmax}_j \left( \left\{ \tfrac{1}{\sqrt{D_h}} (\boldsymbol{k}_{j'}^k)^\top \boldsymbol{q}_i^k \right\}_{j'=1}^{i} \right) \boldsymbol{v}_j^k \in \mathbb{R}^{D_r}$$

denote the output of head $k$. The standard formulation concatenates these outputs,

$$O = [\, O_1^\top,\, O_2^\top,\, \ldots,\, O_K^\top \,]^\top \in \mathbb{R}^{K D_r},$$

and applies a single output projection $\boldsymbol{W}^O \in \mathbb{R}^{K D_r \times D_h}$.

If we partition $\boldsymbol{W}^O$ into head-aligned blocks,

$$\boldsymbol{W}^O = [\, \boldsymbol{W}_1^{O\top},\, \boldsymbol{W}_2^{O\top},\, \ldots,\, \boldsymbol{W}_K^{O\top} \,]^\top, \qquad \boldsymbol{W}_k^O \in \mathbb{R}^{D_r \times D_h},$$

then multiplying out gives

$$\boldsymbol{W}^{O\top} O = [\, \boldsymbol{W}_1^{O\top}, \ldots, \boldsymbol{W}_K^{O\top} \,][\, O_1^\top,\, O_2^\top,\, \ldots,\, O_K^\top \,]^\top = \sum_{k=1}^{K} \boldsymbol{W}_k^{O\top} O_k.$$

Thus, the conventional *concatenation followed by a single projection* is algebraically equivalent to the *sum over head-specific projections* used in our presentation. We adopt the latter form for notational clarity in the CEM formulation.

## A.2    DIAGONAL-PLUS-LOW-RANK PARAMETERISATION

We use the diagonal-plus-low-rank (D+LR) parameterization for both the attention-score computation (in Equation (12)) and the preconditioners (in Equation (13). Here we provide a brief background on this parameterization.

The basic form of a D+LR matrix is often written as

$$W = \text{diag}(d) + UV^\top, \qquad U, V \in \mathbb{R}^{d \times r}, \quad r \ll d.$$

Compared with a pure diagonal parameterization, which cannot express cross-feature interactions, and a pure low-rank parameterization, which only captures interactions within a rank-$r$ subspace, the D+LR form models both aspects. The memory footprint and computational cost are still much smaller compared to the full matrix: applying $W$ requires one diagonal pass and two thin matrix multiplications, with complexity $O(d) + O(dr)$, far smaller than the $O(d^2)$ cost of a dense matrix. D+LR parameterizations has already been widely used in deep learning such as (Gu et al., 2022; Bonnabel et al., 2024).

# B    INCORPORATING POSITIONAL ENCODING INTO CEM ATTENTION

**Positional encoding.**    Standard Transformer architectures such as Llama employ Rotary Position Embeddings (RoPE) (Su et al., 2024) to encode relative position information. Recall from Section 2.1 that in our energy-based formulation, each head $k$ is parameterized by a matrix $\boldsymbol{A}_k$:

$$\boldsymbol{\beta}_{kj} = \boldsymbol{A}_k \boldsymbol{h}_j, \qquad \text{with } \boldsymbol{A}_k \in \mathbb{R}^{D_h \times D_h}.$$

In the simplest case, we adopt a low-rank factorization $\boldsymbol{A}_k = \boldsymbol{W}_k^{Q\top} \boldsymbol{W}_k^K$, so that queries, keys, and values arise as

$$\boldsymbol{q}_i^k = \boldsymbol{W}_k^Q \boldsymbol{h}_i, \quad \boldsymbol{k}_j^k = \boldsymbol{W}_k^K \boldsymbol{h}_j, \quad \boldsymbol{v}_j^k = \boldsymbol{W}_k^V \boldsymbol{h}_j,$$

under the weight-tying constraints $\boldsymbol{W}_k^K = \boldsymbol{W}_k^V$ and $\boldsymbol{W}_k^Q = \boldsymbol{W}_k^O$ (see equation 5). The interaction energy is then defined as

$$\epsilon(\boldsymbol{x}_i \mid \boldsymbol{h}_{1:i}) = -\tau \sum_{k=1}^K \log \sum_{j=1}^i \exp\left(\tfrac{1}{\tau} \boldsymbol{\beta}_{kj}^\top \boldsymbol{x}_i\right),$$

and its gradient update recovers the standard multi-head attention form with weight sharing.

When incorporating RoPE, however, $\boldsymbol{A}_k$ must depend explicitly on both indices $i$ and $j$ through rotation matrices $\boldsymbol{R}(i)$ and $\boldsymbol{R}(j)$:

$$\boldsymbol{A}_k = \boldsymbol{W}_k^{Q\top} \boldsymbol{R}(i)^\top \boldsymbol{R}(j) \boldsymbol{W}_k^K.$$

This makes $\boldsymbol{\beta}_{kj}$ dependent on the query index $i$ as well as $j$, which substantially increases memory costs: the value projection effectively becomes query-dependent.

**Alibi positional encodings.** To mitigate this overhead, we instead adopt **Alibi positional encodings** (Press et al., 2022), which introduce a head-specific bias

$$b_{ijk} = m_k |i - j|$$

directly into the attention scores before the softmax. Concretely, in the unbiased case the score is

$$s_{ijk} = \tfrac{1}{\tau} \boldsymbol{\beta}_{kj}^\top \boldsymbol{x}_i,$$

so with Alibi it becomes

$$s_{ijk} = \tfrac{1}{\tau} \boldsymbol{\beta}_{kj}^\top \boldsymbol{x}_i + b_{ijk},$$

and the normalized weights are

$$\alpha_{ij}^k = \mathrm{softmax}_j\left(\{s_{ij'k}\}_{j'=1}^i\right).$$

The slopes $m_k$ are typically chosen as a geometric sequence, e.g. $m_k = 2^{-k}$. This adds negligible overhead compared to RoPE while still encoding relative bias. In practice, we further include a learnable bias distinguishing self- vs. cross-token attention:

$$b_{ijk} = m_k |i - j| + b_{i=j} + b_{i \neq j}.$$

**Interaction energy with bias.** In the energy formulation, this simply shifts the logits inside the log-sum-exp:

$$\epsilon(\boldsymbol{x}_i \mid \boldsymbol{h}_{1:i}) = -\tau \sum_{k=1}^K \log \sum_{j=1}^i \exp\left(\tfrac{1}{\tau} \boldsymbol{\beta}_{kj}^\top \boldsymbol{x}_i + b_{ijk}\right).$$

The corresponding gradient update is

$$\nabla_{\boldsymbol{x}_i} \epsilon(\boldsymbol{x}_i \mid \boldsymbol{h}_{1:i}) = -\sum_{k=1}^K \sum_{j=1}^i \mathrm{softmax}_j\left(\tfrac{1}{\tau} \boldsymbol{\beta}_{kj}^\top \boldsymbol{x}_i + b_{ijk}\right) \boldsymbol{\beta}_{kj},$$

so $b_{ijk}$ modifies the logits before normalization but leaves the overall gradient structure unchanged.

## C RELATION TO HOPFIELD NETWORKS

Hopfield networks are classical models of associative memory, where stored patterns correspond to attractors of an energy landscape, and the dynamics converge to the attractor most consistent with the initial state. This viewpoint aligns with our interpretation of Transformer layers as energy-minimizing updates: both attention and MLP sublayers can be seen as iterative steps that decrease a suitably defined energy function. We next detail these connections.

**Interaction energy.** Classical Hopfield networks (Hopfield, 1982) store a finite set of patterns $\{\boldsymbol{h}_j\}$ in an energy function of the form

$$\epsilon(\boldsymbol{x}) = -\tfrac{1}{2} \sum_j (\boldsymbol{h}_j^\top \boldsymbol{x})^2,$$

More recent extensions reinterpret Hopfield networks as continuous attractor models, greatly expanding their representational capacity. For instance, dense associative memories (Krotov & Hopfield, 2016) and modern Hopfield networks (Ramsauer et al., 2021) introduce an energy of the log-sum-exp form,

$$\epsilon(\boldsymbol{x}) = -\tau \log \sum_j \exp\!\Big(\tfrac{1}{\tau} \boldsymbol{h}_j^\top \boldsymbol{x}\Big),$$

which is convex in $\boldsymbol{x}$ and whose fixed-point updates under the concave-convex procedure (CCCP) (Yuille & Rangarajan, 2003) yield

$$\boldsymbol{x}' = \sum_j \operatorname{softmax}_j\Big(\tfrac{1}{\tau} \boldsymbol{h}_j^\top \boldsymbol{x}\Big)\, \boldsymbol{h}_j,$$

exactly the update rule underlying the attention mechanism. This connection underlies the interpretation of attention as a form of fast Hopfield retrieval.

**Our perspective.** We depart from the setup of modern Hopfield networks in three important ways. First, instead of computing fixed points via iterative CCCP updates (Yuille & Rangarajan, 2003), we interpret each Transformer sublayer as performing a *single gradient step* on an energy function. Second, in our formulation the query and key projection matrices are embedded directly in the energy, which causes them to reappear as the output–value projections in the gradient update—naturally yielding the tied $\boldsymbol{W}_Q, \boldsymbol{W}_K$ and $\boldsymbol{W}_O, \boldsymbol{W}_V$ structure of attention. Finally, while Ramsauer et al. (2021) introduce novel Hopfield layers and evaluate them on associative-memory benchmarks, our framework treats standard Transformer layers themselves as energy-based updates, and we demonstrate that this perspective leads to principled extensions and improvements for text modeling tasks.

**Element-wise energy.** The element-wise energy leading to gated MLPs has a less direct connection. Optimization via CCCP is possible only when using a convex form. We briefly experimented with models using energies of the form

$$\xi(\boldsymbol{x}_i \mid \boldsymbol{h}_i) = -|\boldsymbol{\gamma}_i|^\top \phi\big(\operatorname{diag}(\operatorname{sign}(\boldsymbol{\gamma}_i))\, \boldsymbol{V} \boldsymbol{x}_i\big), \qquad \boldsymbol{\gamma}_i = \boldsymbol{W} \boldsymbol{h}_i,$$

with $\phi$ a convex nonlinearity, so that the energy is convex in $\boldsymbol{x}$. The gradient of this energy form is

$$-\boldsymbol{V}^\top \big(\boldsymbol{\gamma}_i \circ \phi'(\operatorname{diag}(\operatorname{sign}(\boldsymbol{\gamma}_i))\boldsymbol{V} \boldsymbol{x}_i)\big)$$

We used a straight-through estimator to deal with the sign nonlinearity. We found that these models successfully trained, but with worse performance than ignoring the sign. Unlike the interaction energy, the link to memory association here is unclear, as are the corresponding convergence guarantees and capacity limits.

# D    EXPERIMENTAL DETAILS

## D.1    MODEL ARCHITECTURES

We evaluate CEM-based architectures across multiple model scales ranging from 86M to 162M parameters. All models follow the Llama architecture as baseline with modifications for CEM components. Table 1 summarizes the architectural details for each model size.

## D.2    DATASET AND PREPROCESSING

**Dataset.** We use a subset of SlimPajama-627B (Soboleva et al., 2023), a cleaned and deduplicated variant of RedPajama comprising approximately 627 billion tokens drawn from Common-Crawl, C4, GitHub, books, arXiv, Wikipedia, and StackExchange. The dataset is accessed via `gmongaras/SlimPajama-627B_Reupload` on Hugging Face.

Table 1: Model architecture configurations for different parameter counts. All models use a vocabulary size of 32,000 tokens.

| Configuration | 86M | 108M | 134M | 162M |
|---|---|---|---|---|
| Model dimension ($d_h$) | 672 | 672 | 768 | 864 |
| Number of layers | 8 | 12 | 12 | 12 |
| Number of heads | 8 | 12 | 12 | 12 |
| MLP intermediate dimension | 1792 | 1792 | 2048 | 2304 |
| Context length | 2048 | 2048 | 2048 | 2048 |

**Tokenization.** We employ the `LlamaTokenizerFast` with a vocabulary size of 32,000 tokens.

**Data processing.** Documents are concatenated and split into fixed-length sequences of 2048 tokens, with no padding applied.

### D.3 TRAINING CONFIGURATION

Training hyperparameters are summarized in Table 2. We follow Chinchilla-optimal compute allocation (Hoffmann et al., 2022) for determining the number of training tokens for each model size.

Table 2: Training hyperparameters for CEM models and Llama baselines.

| Hyperparameter | CEM models | Llama baseline |
|---|---|---|
| Optimizer | | AdamW |
| Learning rate | | 0.002 |
| $\beta_1$ | | 0.9 |
| $\beta_2$ | | 0.95 |
| $\epsilon$ | | 1e-9 |
| Weight decay | | 0.1 |
| Gradient clipping | | 1.0 |
| LR schedule | | Cosine |
| Warmup steps | | 5% of total |
| Final LR factor | | 0.1 |
| Batch size (per GPU) | | 8 |
| Gradient accumulation | | 4 |
| Effective batch size | | 128 |
| Precision | | bf16-mixed |

### D.4 INITIALISATION OF PRECONDITIONERS

In Section 2.3, we introduce a trainable diagonal-plus-low-rank preconditioner of the form

$$\boldsymbol{P} = \mathrm{diag}\big(\mathrm{softplus}(\boldsymbol{d})\big) + \boldsymbol{U}\boldsymbol{V}^\top + \boldsymbol{V}\boldsymbol{U}^\top.$$

with $\boldsymbol{d} \in \mathbb{R}^{D_h}$ and $\boldsymbol{U}, \boldsymbol{V} \in \mathbb{R}^{D_h \times R}$, where $R \ll D_h$. Following Hu et al. (2022), we initialize $\boldsymbol{U}$ from a normal distribution ($\sigma = 0.02$)and set $\boldsymbol{V}$ to zeros. For the diagonal term, we parameterize

$$\boldsymbol{d} = \sqrt{D_h}\,\boldsymbol{p},$$

where $\boldsymbol{p}$ is initialized to $1/\sqrt{D_h}$. This ensures that $\boldsymbol{d}$ starts at 1, but still yielding an appropriate effective gradient step size.

To keep the preconditioners lightweight, we set $R = 4$ for attention modules and $R = 16$ for MLPs. In the diagonal-only case, the preconditioner reduces to

$$\boldsymbol{P} = \mathrm{diag}\big(\mathrm{softplus}(\boldsymbol{d})\big).$$

### D.5 Compute resources

All experiments were conducted on a cluster of $8\times$ NVIDIA A100 GPUs (40GB memory each). Training time per model scales with size: the smallest models ($\sim$86M parameters) require about $8 \times 2$ GPU-hours, while the largest models we tested ($\sim$162M parameters) require about $8 \times 18$ GPU-hours. End-to-end reproduction of all results in this paper would therefore require on the order of 10,000 GPU-hours.

## E Additional results

**Recursive updates in MHA** Similar to our analysis of MLP recursion Figure 3c, we examine recursive updates in attention layers (Figure 4). As with MLPs, naive reuse of the same MHA block offers no benefit and can even degrade performance in the case of MHA. In contrast, within-layer recursion in CEM attention yields clear and consistent perplexity improvements.

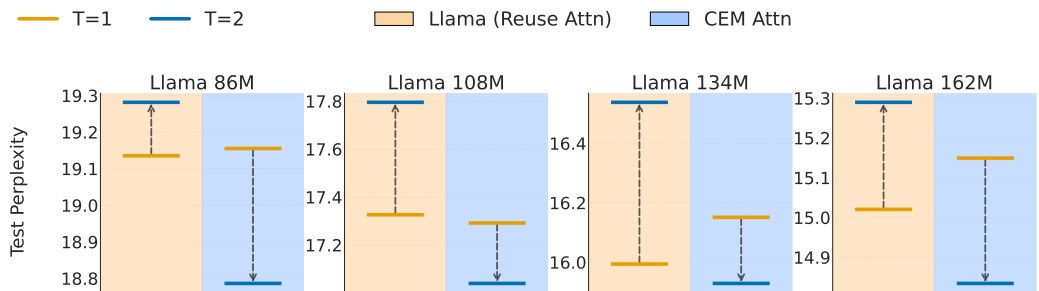

Figure 4: Within-layer recursion vs. plain layer reuse in MHAs. We compare the performance gains of increasing recursion from $T = 1$ (orange) to $T = 2$ (blue), under three settings: Plain layer reuse (light orange area), dimension-matched CEM MHA (blue area).

**Study recursion on synthetic data** To better isolate and understand the intrinsic behaviour of the recursion, we also examine it in a controlled and computationally lightweight setting using Gaussian-process generated data and fit with our recursive CEM MLP. The results in Table 3, illustrate that additional recursive steps generally improve performance, though the gains are not strictly monotonic.

**Akima interpolation with larger models** We scale our models to 256M parameters, and results analogous to Figure 3a are shown in Figure 5. We find that CEM models, despite having fewer parameters, continue to outperform Llama models at this scale. Scaling to substantially larger sizes would require significantly more computational resources, and we leave this for future work.

## F LLM Usage Statement

We used ChatGPT-5 to assist with paraphrasing, text editing, and proofreading. We also used Chat-GPT to help search for and discover relevant related work. All conceptual development, technical contributions, experiments, and analysis were carried out by the authors.

Table 3: Model size, compute, and RMSE (mean ± std) on synthetic data sampled from Gaussian processes with 10 input dimensions. Here the latent state dimension is set equal to the model dimension. Values are averaged over repeated runs, and the lowest (best) train and test RMSE for each kernel are highlighted in **bold**. FLOPs are reported per forward pass. (Abbreviations: $K = 10^3$, $M = 10^6$.)

| MODEL | PARAMETERS | FLOPs | KERNEL | TRAIN RMSE | TEST RMSE |
|-------|-----------|-------|--------|-----------|-----------|
| Plain | 33.98K | 6.76M | RBF | $0.0139 \pm 0.0022$ | $0.0158 \pm 0.0024$ |
| Gated | 33.94K | 6.73M | RBF | $0.0109 \pm 0.0014$ | $0.0122 \pm 0.0015$ |
| CEM-T1 | 22.66K | 6.73M | RBF | $0.0102 \pm 0.0010$ | $0.0119 \pm 0.0011$ |
| CEM-T2 | 22.66K | 11.08M | RBF | $0.0074 \pm 0.0008$ | $0.0092 \pm 0.0008$ |
| CEM-T4 | 22.66K | 19.79M | RBF | $0.0068 \pm 0.0006$ | $\mathbf{0.0086 \pm 0.0007}$ |
| CEM-T8 | 22.66K | 37.20M | RBF | $\mathbf{0.0066 \pm 0.0008}$ | $0.0088 \pm 0.0012$ |
| Plain | 33.98K | 6.76M | Matern | $0.2129 \pm 0.0205$ | $0.2308 \pm 0.0237$ |
| Gated | 33.94K | 6.73M | Matern | $0.1903 \pm 0.0205$ | $0.2162 \pm 0.0253$ |
| CEM-T1 | 22.66K | 6.73M | Matern | $0.1836 \pm 0.0199$ | $0.2194 \pm 0.0267$ |
| CEM-T2 | 22.66K | 11.08M | Matern | $\mathbf{0.1623 \pm 0.0170}$ | $0.2166 \pm 0.0274$ |
| CEM-T4 | 22.66K | 19.79M | Matern | $0.1742 \pm 0.0206$ | $0.2168 \pm 0.0271$ |
| CEM-T8 | 22.66K | 37.20M | Matern | $0.1676 \pm 0.0174$ | $\mathbf{0.2161 \pm 0.0248}$ |
| Plain | 33.98K | 6.76M | Periodic | $0.0557 \pm 0.0056$ | $0.0614 \pm 0.0058$ |
| Gated | 33.94K | 6.73M | Periodic | $0.0386 \pm 0.0036$ | $0.0445 \pm 0.0043$ |
| CEM-T1 | 22.66K | 6.73M | Periodic | $0.0352 \pm 0.0036$ | $0.0421 \pm 0.0044$ |
| CEM-T2 | 22.66K | 11.08M | Periodic | $0.0240 \pm 0.0022$ | $0.0342 \pm 0.0033$ |
| CEM-T4 | 22.66K | 19.79M | Periodic | $\mathbf{0.0238 \pm 0.0029}$ | $\mathbf{0.0340 \pm 0.0042}$ |
| CEM-T8 | 22.66K | 37.20M | Periodic | $0.0242 \pm 0.0037$ | $0.0344 \pm 0.0053$ |
| Plain | 33.98K | 6.76M | Rational Quadratic | $0.0791 \pm 0.0109$ | $0.0885 \pm 0.0108$ |
| Gated | 33.94K | 6.73M | Rational Quadratic | $0.0599 \pm 0.0070$ | $0.0715 \pm 0.0075$ |
| CEM-T1 | 22.66K | 6.73M | Rational Quadratic | $0.0578 \pm 0.0072$ | $0.0709 \pm 0.0082$ |
| CEM-T2 | 22.66K | 11.08M | Rational Quadratic | $0.0409 \pm 0.0039$ | $\mathbf{0.0596 \pm 0.0068}$ |
| CEM-T4 | 22.66K | 19.79M | Rational Quadratic | $0.0417 \pm 0.0033$ | $0.0606 \pm 0.0067$ |
| CEM-T8 | 22.66K | 37.20M | Rational Quadratic | $\mathbf{0.0406 \pm 0.0035}$ | $0.0606 \pm 0.0060$ |
| Plain | 33.98K | 6.76M | Non-Stationary | $0.0385 \pm 0.0052$ | $0.0426 \pm 0.0062$ |
| Gated | 33.94K | 6.73M | Non-Stationary | $0.0315 \pm 0.0030$ | $0.0358 \pm 0.0039$ |
| CEM-T1 | 22.66K | 6.73M | Non-Stationary | $0.0305 \pm 0.0044$ | $0.0356 \pm 0.0051$ |
| CEM-T2 | 22.66K | 11.08M | Non-Stationary | $0.0257 \pm 0.0027$ | $0.0330 \pm 0.0042$ |
| CEM-T4 | 22.66K | 19.79M | Non-Stationary | $0.0267 \pm 0.0030$ | $0.0330 \pm 0.0043$ |
| CEM-T8 | 22.66K | 37.20M | Non-Stationary | $\mathbf{0.0243 \pm 0.0030}$ | $\mathbf{0.0317 \pm 0.0043}$ |

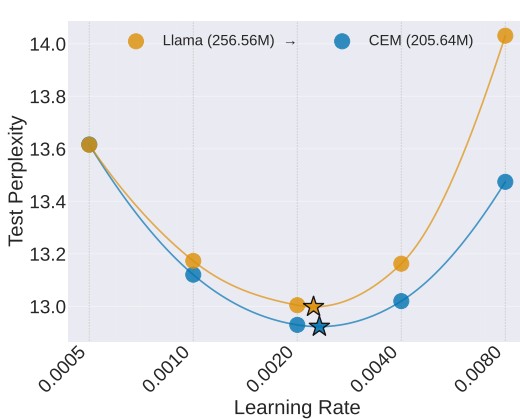

Figure 5: Additional results for optimal learning-rate estimation via Akima interpolation. Orange curves denote baseline Llama models; blue curves denote CEM models ($T = 2$) with both MHA and MLP replaced. Marker shapes indicate model size, and star markers denote interpolated optima from five data points. CEM models use roughly half the MHA parameters and one-third the MLP parameters per layer, with two additional layers to offset the larger parameter reduction for this size, and are trained under the same token budget (Chinchilla-optimal for Llama).

