# OpenReview forum: "Recasting Transformer Layers as Energy Models"
_ICLR.cc/2026/Conference — ICLR 2026 Conference Desk Rejected Submission_

### Official Review · Reviewer_5RHo · 2025-10-23

**Soundness:** 3
**Presentation:** 3
**Contribution:** 3
**Rating:** 6
**Confidence:** 4

**Summary:**

This paper reinterprets Transformer layers from an optimization perspective. It shows that both multi-head attention (MHA) and gated MLP blocks can be expressed as single gradient descent steps on simple energy functions: a log-sum-exp interaction energy for attention and an elementwise energy for MLP. This provides a unified view of Transformer blocks as implicit optimizers. Based on this view, the authors propose architectural variants such as diagonal + low-rank parameterization, lightweight preconditioners, and multiple inner steps (T > 1), achieving improved parameter efficiency and perplexity on language modeling benchmarks. The framework is closely related to Hopfield network interpretations but extends them by covering both attention and MLP.

**Strengths:**

**Unified Framework**: Provides a single energy-based optimization perspective that unifies MHA and MLP within the Transformer block.

**Actionable Insights**: Goes beyond theoretical interpretation by proposing architectural modifications directly derived from the framework (diagonal + low-rank parameterization, preconditioners, T-step updates).

**Empirical Gains**: Demonstrates improved parameter efficiency and lower perplexity on language modeling benchmarks.

**Weaknesses:**

**Overlap with prior work**: The paper’s framing is conceptually close to Hopfield-based interpretations of attention. The novelty relies heavily on unifying MHA and MLP.

**Lack of formal guarantees**: There is no analysis of convergence, stability, or approximation error when T > 1. The theory remains at an intuitive level.

**Experimental scale limitations**: Experiments are restricted to mid-sized models. No large-scale results (e.g., 7B+ parameters) are provided. It is unclear how the method scales.

**Reproducibility**: No implementation or code is provided, which makes it difficult to fully verify the empirical claims.

**Questions:**

Minor Errors

Line 25:  pre-conditioner ->  preconditioner

Line 26: inter-token-distances -> inter-token distances

Line 20: weights sharing -> weight-sharing

Line 39: transformers continues -> transformers continue

Line 177: Comparing equations equation 6 and equation 10 -> Comparing Equation (6) and Equation (10)

Line 188: Diagonal-plus-low-rank parameterisation In Section 2.1 -> Diagonal-plus-low-rank parameterisation. In Section 2.1

Line 269: netowrks -> networks

Line 742: to that the energy -> so that the energy

Line 755: components.Table 1 -> components. Table 1

The term Transformer is inconsistently capitalized throughout the manuscript (e.g., “Transformer” vs. “transformer”).

Line 253: \mathbf{k}_{1:i} -> \mathbf{k}^k_{1:i}, \mathbf{v}_{1:i} -> \mathbf{v}^k_{1:i}

---

> ### Author Response · Authors · 2025-11-25
> **Author Response to 5RHo**
>
> > “Overlap with prior work: The paper’s framing is conceptually close to Hopfield-based interpretations of attention. The novelty relies heavily on unifying MHA and MLP.”
>
> **Response:**
> While the Hopfield network paper also provides a pattern retrieval view of MHA, our framework goes further by offering a unified energy formulation for both multi-head attention and gated MLPs.
>
> In addition, the energy-minimization perspective naturally motivates several architectural extensions that we validate empirically, including (i) multi-step updates, (ii) per-head preconditioning, and (iii) diagonal distance matrices. Each of these consistently improves language-modeling performance in our experiments.
>
> Finally, our formulation induces weight sharing in both MHA and MLP. In particular, a CEM block shares the key/value and value/output projection weights, as well as the up/down projection weights. This is an aspect not discussed in the Hopfield paper. With appropriate kernel implementations, these shared structures has the potential to translate into practical efficiency gains.
>
>
> > “There is no analysis of convergence, stability, or approximation error when T > 1. The theory remains at an intuitive level.”
>
> **Response:**
> We appreciate the reviewer’s comment that a deeper theoretical analysis of convergence, stability, or approximation error for $𝑇>1$, would strengthen the work. Our primary goal in this paper is to introduce the CEM framework and establish its viability, and we therefore focus on empirically evaluating the within-layer recursion, an example of the layer extensions motivated by our energy-based perspective, rather than developing a full convergence theory.
>
> We agree that a complete convergence or stability analysis for arbitrary $T$ would be valuable. However, such a theory requires analyzing learned, data-dependent updates and is beyond the scope of the current work. We view this as an important direction for future work.
>
> > “Experiments are restricted to mid-sized models. No large-scale results (e.g., 7B+ parameters) are provided. It is unclear how the method scales.”
>
> **Response:**
> Our goal in this paper is to introduce and validate the CEM framework, and we therefore focus empirically on 80M–160M models to demonstrate parameter efficiency and competitive performance at this scale. Training 7B+ models requires substantial compute and engineering resources, which is beyond the scope of the current work.
>
> In light of the reviewer’s suggestion, we expanded our experiments during the rebuttal period and now include a 256M-scale result in the revised manuscript (Appendix E, Fig. 5). Our 205M CEM model outperforms the 256M Llama baseline in perplexity, providing additional evidence that the approach continues to behave well as model size increases.
>
> While full 7B-scale validation would indeed strengthen the study, pre-training such models from scratch is prohibitively resource-intensive for us at this stage. The 256M experiments reflect the largest extension feasible within our computational budget
>
> > “No implementation or code is provided, which makes it difficult to fully verify the empirical claims.”
>
> **Response:**
> We are in the process of releasing a full implementation, including model configurations, and all training scripts. We intend to make these publicly available soon.
>
> > The list of editorial issues/minor issues
>
> **Response:**
> Thanks a lot for the detailed list. We have now corrected all these editorial issues in our revised version uploaded.

---

### Official Review · Reviewer_i5ES · 2025-10-29

**Soundness:** 3
**Presentation:** 3
**Contribution:** 3
**Rating:** 6
**Confidence:** 2

**Summary:**

This paper introduces a framework called Causal Energy Minimization (CEM), which reinterprets each component of Transformer layers (such as multi-head attention mechanism MHA and gated multilayer perceptron gated MLP) as an optimization algorithm for solving energy minimization problems with causal structure. This framework separates the mathematical semantics of the layer (the defined energy function) from its numerical realization (the optimization algorithm used to minimize the energy), thereby providing a unified perspective for understanding and designing Transformer layers. The paper reexamines the Transformer through the lens of energy models, offering a principled and extensible viewpoint.

**Strengths:**

1. The work demonstrates originality in its unique framework that combines energy-based models (EBMs) with Transformer layers. The framework innovatively formalizes each Transformer layer as a causal energy minimization problem and derives MHA and gated MLP as single-step gradient updates on specific energy functions.
2. The work is well-structured, with overall writing that is fluent and suitable for AI/ML readers.
3. The work holds significance by providing a principled framework for Transformer architecture, potentially guiding future innovations. In the LLM era, architecture design still relies on empirical approaches (such as Llama-style), and CEM bridges optimization theory with sequence modeling, which may inspire more efficient and interpretable models.

**Weaknesses:**

The main weaknesses of the paper lie in the insufficient scale and breadth of the experiments.

1. The experiments are limited to small-scale models (with a maximum of 160M parameters) and a single dataset (SlimPajama), lacking validation on larger scales (such as 1B+ parameters) or diverse tasks. For example, Figure 2 shows that CEM performs effectively on small models, but in large models, weight sharing may lead to insufficient expressiveness, as larger models rely on more parameters to capture complex patterns.

**Questions:**

Can larger models (such as 1B+ parameters) be included? If the authors can provide preliminary results showing CEM's performance on larger scales, this would greatly enhance my confidence, as small-model results may not represent large-model behavior.

---

> ### Author Response · Authors · 2025-11-25
> **Author Response to Reviewer i5ES**
>
> We thank the reviewer for the thoughtful evaluation and feedback. Our responses are as follows:
>
> > "The experiments are limited to small-scale models (with a maximum of 160M parameters) and a single dataset (SlimPajama), lacking validation on larger scales (such as 1B+ parameters) or diverse tasks."
>
> **Response:**
> We appreciate the reviewer’s point that larger-scale validation would further strengthen the empirical evidence. Our primary goal in this work is to introduce and validate the CEM framework, and we therefore focus on 80M–160M–scale experiments to demonstrate parameter efficiency and competitive performance with our layer design. Scaling to 1B+ models requires substantial compute and engineering effort, which is beyond the scope of the current paper.
>
> In response to this helpful suggestion, we have expanded our experiments and now include 256M-scale results in the revised manuscript (Appendix E, Fig. 5). These results show that our 205M CEM model achieves lower perplexity than the 256M Llama baseline under Akima interpolation. We hope this provides additional confidence in the framework.
>
> We fully agree on the value of further scaling, but pre-training larger models from scratch is highly resource-intensive. The 256M-scale experiments represent the largest realistic extension we can provide within our current computational budget.

---

### Official Review · Reviewer_jswD · 2025-10-30

**Soundness:** 3
**Presentation:** 3
**Contribution:** 3
**Rating:** 6
**Confidence:** 3

**Summary:**

This paper introduces Causal Energy Minimization (CEM), a novel framework that reinterprets Transformer layers as algorithms for solving energy minimization problems with causal structure. The key insight is that multi-head attention (MHA) emerges naturally as a gradient descent step on an interaction energy function, while gated MLPs correspond to gradient updates on element-wise energies. This energy-based perspective reveals that standard Transformer components can be derived under weight-sharing constraints: specifically, $W_K = W_V$ and $W_Q = W_O$ in attention, and $W^{d⊤} = W^u = V$ in MLPs.

CEM provides a principled framework for understanding why current architectures work and offers systematic guidance for designing new components.

**Strengths:**

Building on the energy optimization perspective, the authors propose three principled enhancements:
- (1) Diagonal-plus-low-rank parameterization for attention matrices ( $A_k = \text{diag}(d_k) + W_k^{Q⊤}W_k^K$ ) to capture richer token interactions.
- (2) Learned lightweight preconditioners in diagonal-plus-low-rank form to accelerate convergence.
- (3) Within-layer recursive updates (multiple gradient steps, T=2) that improve optimization without adding parameters.

Experiments on SlimPajama with models ranging from 86M to 162M parameters demonstrate that CEM layers serve as effective parameter-efficient replacements for standard Transformer components: CEM attention uses approximately half the parameters with minimal performance loss, while recursive updates (T=2) with preconditioners can even outperform Llama baselines. Full end-to-end CEM Transformers achieve slightly better perplexity at optimal learning rates while using considerably fewer parameters. The framework positions CEM as a blueprint for principled architecture design, though the authors acknowledge that custom kernels and hardware co-design remain important future work to fully realize efficiency gains in practice.

**Weaknesses:**

One of my primary concerns is that while the energy model framework elegantly echoes the mathematical form of attention in Transformer architectures, the connection to language modeling tasks appears fundamentally superficial. I acknowledge that numerous prior works have reinterpreted attention through various theoretical lenses—such as kernel regression, implicit gradient descent, or associative memory retrieval—but the question remains: **why should we view Transformers specifically as energy models for language modeling?** The paper lacks language-modeling-specific analysis that would justify this perspective. For instance, there is no investigation into how the energy minimization interpretation improves training stability, gradient flow dynamics, loss landscape geometry, or generalization properties in the context of next-token prediction. Without such task-relevant analysis, the energy model theory risks being merely a post-hoc mathematical reformulation rather than providing actionable insights. Does minimizing the proposed interaction energy $\epsilon(x_i|h_{1:i})$ correspond to any meaningful linguistic or semantic objective? How does this relate to the cross-entropy loss used in language model training? These connections are not established.

Furthermore, I have significant concerns regarding the practical viability of the recursive looping mechanism in CEM-enhanced multi-head attention. The paper reports that recursion becomes unstable for $T \geq 3$ (mentioned on page 11: "we only study recursive steps until T=2 because T≥3 has unstable performance and leads to OOM for larger model sizes"). This suggests the approach has inherent scalability limitations. **What is the fundamental cause of this instability?** Is it due to gradient explosion/vanishing through the recursive loops, insufficient regularization, or the energy landscape becoming pathological with deeper unrolling? The lack of ablation studies on this instability is concerning.

Additionally, the authors do not compare their within-layer recursion to simpler alternatives like adding more Transformer layers with the same parameter budget, which would clarify whether the recursion provides unique benefits beyond simply increasing model depth. The trade-off between the theoretical appeal of iterative energy minimization and the practical constraint of T≤2 undermines the motivation for this design choice.

**Questions:**

Could you provide empirical evidence demonstrating that the energy minimization perspective offers practical benefits specific to language modeling tasks?*

For example:
- How does the energy landscape correlate with language modeling loss during training?
- Does lower energy at intermediate layers correspond to better next-token prediction accuracy?
- Can you show that models trained with explicit energy minimization objectives (e.g., auxiliary losses based on $\epsilon(x_i|h_{1:i})$) improve perplexity, training stability, or sample efficiency compared to standard cross-entropy training?

Without such task-specific validation, it remains unclear whether the energy model framework is more than a mathematical reinterpretation that happens to fit the Transformer equations.

Is the instability due to gradient pathologies (vanishing/exploding), optimization challenges in the energy landscape, or memory constraints?

Have you tried stabilization techniques such as gradient clipping per recursive step, layer normalization within the loop, adaptive step sizes $\eta^{(t)}$, or energy-based early stopping criteria?

Can you provide gradient norm statistics, loss curves, or energy trajectories that reveal when and why the recursion diverges?

Test perplexity at matched FLOPs? and Training throughput and memory usage?

p.s. overall I think this paper is quite interesting in terms of the parameter reduction and looping mechanism.

---

> ### Author Response · Authors · 2025-11-21
> **Author Response to Reviewer jswD (Part 1)**
>
> Thank you very much for the constructive and thoughtful comments. We address each major point in detail below:
>
> > "Why should we view Transformers specifically as energy models for language modeling? The paper lacks language-modeling-specific analysis that would justify this perspective. For instance, there is no investigation into how the energy minimization interpretation improves training stability, gradient flow dynamics, loss landscape geometry, or generalization properties in the context of next-token prediction. "
>
> > "Could you provide empirical evidence demonstrating that the energy minimization perspective offers practical benefits specific to language modeling tasks?"
>
> **Response:**
>
> **Why talking energy model perspective:**
>
> Our framing of Transformers as energy models is primarily motivated by an information-retrieval interpretation of transformer layers. In multi-head attention, the query retrieves information from key–value pairs across the sequence dimension, where keys and values are derived from the corresponding tokens. In MLPs, the input similarly retrieves information from learned key–value pairs (i.e., rows of the weight matrices) across the hidden dimension. This perspective is consistent with *[Hopfield Networks is All You Need](https://arxiv.org/abs/2008.02217)*, which established the connection between multi-head attention and associative-memory retrieval.
>
> For our gated MLP, we can rewrite Eq. (7) as
> $$
> E(x \mid h_i) = \sum_{j=1}^{D_m} -\alpha_{ij}\sigma(v_j^\top x),
> $$
>
> which matches the form of Eq. (9) in *Hopfield Networks is All You Need*, where each $v_j$ can be interpreted as a stored pattern in an associative memory. Under this view, the gates $\alpha_{ij} = w_j^\top h_i$ act as input-dependent coefficients that modulate the contribution of each pattern vector.
>
> This unified retrieval viewpoint naturally leads to an energy-minimization formulation. Investigating how these stored patterns relate to semantic structure in language modeling is an interesting but non-trivial direction, essentially a separate line of work on transformer interpretability. There is already empirical evidence suggesting this, for example:
>
> * *[Transformer Feed-Forward Layers Are Key-Value Memories](https://arxiv.org/abs/2012.14913)*
> * *[Transformer Key-Value Memories Are Nearly as Interpretable as Sparse Autoencoders](https://arxiv.org/abs/2510.22332)*
> * and the theoretical insights from *[Hopfield Networks is All You Need](https://arxiv.org/abs/2008.02217)*.
>
>
> The aspects mentioned by the reviewer, training stability, gradient flow, loss-landscape geometry, and generalization, are indeed interesting, but they are not the primary motivation of this work. To avoid possible misunderstandings, we would like to emphasize that we interpret the forward pass of transformer layers as performing an energy-minimization update. This is distinct from viewing model training (i.e., optimization of the cross-entropy loss) as an energy-based learning problem. As such, our energy function is used for layer design motivated by the pattern retieval perspective, not as a surrogate for the training loss, and we do not claim direct implications for the optimization properties mentioned above.
>
>
> **CEM has practical benefits:**
>
> 1. Our results show that natural extensions of transformer layers inspired by the energy-minimization perspective consistently lead to improvements in language modeling performance. This includes (i) multi-step updates, (ii) preconditioning, (iii) weight sharing (at least on a perplexity-per-parameter basis), and (iv) diagonal distance matrices.
>
> 2. Many real-world applications require language models to operate under memory constraints. By interpreting transformer updates as energy-minimization steps, CEM provides a principled way to design lightweight preconditioners and recursive updates that maintain competitive performance with fewer parameters. This makes the framework particularly well-suited for resource-limited settings, where static memory footprint (proportional to parameter count) is often the dominant bottleneck.
>
> 3. Modern accelerators are often bottlenecked by memory bandwidth rather than compute, especially during autoregressive decoding. The energy-minimization view naturally encourages weight sharing (e.g., shared up/down projections in MLPs and shared key/value projections in attention), which may reduce memory movement and improve on-chip data reuse. Although we do not pursue wall-clock speedups in this work, achieving them would primarily require specialized kernel implementations. With such kernels, the parameter efficiency of CEM-based layers could potentially translate into bandwidth savings and modest runtime improvements during decoding. We offer this only as a possible opportunity suggested by the framework, not as a demonstrated result.
>
> (edit: fixed typo in equation)

---

> > ### Author Response · Authors · 2025-11-21
> > **Author Response to Reviewer jswD (Part 2)**
> >
> > > "The paper reports that recursion becomes unstable for
> >  (mentioned on page 11: "we only study recursive steps until T=2 because T≥3 has unstable performance and leads to OOM for larger model sizes"). This suggests the approach has inherent scalability limitations. What is the fundamental cause of this instability?"
> >
> > > "Is the instability due to gradient pathologies (vanishing/exploding), optimization challenges in the energy landscape, or memory constraints?"
> >
> >
> > **Response:**
> > We acknowledge that we do not have a fully robust recipe for stable recursion at larger $T$. However, we do not regard this as a fundamental limitation of the recursion mechanism itself; rather, it reflects the need for more refined initialization schemes, parameterizations, and step-size schedules, which we haven't find a satisfactory solution yet, so we only show the consistent improvement at $T=2$ in our manuscript.
> >
> > We believe the instability at larger $T$ may be similar to the well-known challenges of training deep recurrent networks with backpropagation through time, particularly when such recursive steps are embedded within an already deep transformer architecture. It also makes the dynamics increasingly sensitive to initialization scales and other hyperparameters.
> >
> > To better isolate and understand the intrinsic behaviour of the recursion, we examine it in a controlled and computationally lightweight setting using Gaussian-process generated data and fit with our recursive CEM MLP. The results, summarized below and included in the revised manuscript, illustrate that additional recursive steps generally improve performance, though the gains are not strictly monotonic, likely reflecting the same trainability considerations discussed above.
> >
> > Table: RMSE (mean $\pm$ std) on synthetic data sampled from Gaussian processes with 10 input dimensions and different kernels.
> > Here the optimized state $x$ is set to have the same dimension as the input state $h$.
> > Values are averaged over repeated runs, and the lowest (best) train and test RMSE for each kernel are highlighted in bold.
> > FLOPs are reported per forward pass.
> >
> > | Model   | Kernel              | Train RMSE              | Test RMSE               |
> > |---------|----------------------|-------------------------|--------------------------|
> > | Plain   | RBF                  | 0.0139 ± 0.0022         | 0.0158 ± 0.0024          |
> > | Gated   | RBF                  | 0.0109 ± 0.0014         | 0.0122 ± 0.0015          |
> > | CEM-T1  | RBF                  | 0.0102 ± 0.0010         | 0.0119 ± 0.0011          |
> > | CEM-T2  | RBF                  | 0.0074 ± 0.0008         | 0.0092 ± 0.0008          |
> > | CEM-T4  | RBF                  | 0.0068 ± 0.0006         | **0.0086 ± 0.0007**      |
> > | CEM-T8  | RBF                  | **0.0066 ± 0.0008**     | 0.0088 ± 0.0012          |
> > | Plain   | Matern               | 0.2129 ± 0.0205         | 0.2308 ± 0.0237          |
> > | Gated   | Matern               | 0.1903 ± 0.0205         | 0.2162 ± 0.0253          |
> > | CEM-T1  | Matern               | 0.1836 ± 0.0199         | 0.2194 ± 0.0267          |
> > | CEM-T2  | Matern               | **0.1623 ± 0.0170**     | 0.2166 ± 0.0274          |
> > | CEM-T4  | Matern               | 0.1742 ± 0.0206         | 0.2168 ± 0.0271          |
> > | CEM-T8  | Matern               | 0.1676 ± 0.0174         | **0.2161 ± 0.0248**      |
> > | Plain   | Periodic             | 0.0557 ± 0.0056         | 0.0614 ± 0.0058          |
> > | Gated   | Periodic             | 0.0386 ± 0.0036         | 0.0445 ± 0.0043          |
> > | CEM-T1  | Periodic             | 0.0352 ± 0.0036         | 0.0421 ± 0.0044          |
> > | CEM-T2  | Periodic             | 0.0240 ± 0.0022         | 0.0342 ± 0.0033          |
> > | CEM-T4  | Periodic             | **0.0238 ± 0.0029**     | **0.0340 ± 0.0042**      |
> > | CEM-T8  | Periodic             | 0.0242 ± 0.0037         | 0.0344 ± 0.0053          |
> > | Plain   | Rational Quadratic   | 0.0791 ± 0.0109         | 0.0885 ± 0.0108          |
> > | Gated   | Rational Quadratic   | 0.0599 ± 0.0070         | 0.0715 ± 0.0075          |
> > | CEM-T1  | Rational Quadratic   | 0.0578 ± 0.0072         | 0.0709 ± 0.0082          |
> > | CEM-T2  | Rational Quadratic   | 0.0409 ± 0.0039         | **0.0596 ± 0.0068**      |
> > | CEM-T4  | Rational Quadratic   | 0.0417 ± 0.0033         | 0.0606 ± 0.0067          |
> > | CEM-T8  | Rational Quadratic   | **0.0406 ± 0.0035**     | 0.0606 ± 0.0060          |
> > | Plain   | Non-Stationary       | 0.0385 ± 0.0052         | 0.0426 ± 0.0062          |
> > | Gated   | Non-Stationary       | 0.0315 ± 0.0030         | 0.0358 ± 0.0039          |
> > | CEM-T1  | Non-Stationary       | 0.0305 ± 0.0044         | 0.0356 ± 0.0051          |
> > | CEM-T2  | Non-Stationary       | 0.0257 ± 0.0027         | 0.0330 ± 0.0042          |
> > | CEM-T4  | Non-Stationary       | 0.0267 ± 0.0030         | 0.0330 ± 0.0043          |
> > | CEM-T8  | Non-Stationary       | **0.0243 ± 0.0030**     | **0.0317 ± 0.0043**      |

---

> > > ### Author Response · Authors · 2025-11-21
> > > **Author Response to Reviewer jswD (Part 3)**
> > >
> > > > "Additionally, the authors do not compare their within-layer recursion to simpler alternatives like adding more Transformer layers with the same parameter budget, which would clarify whether the recursion provides unique benefits beyond simply increasing model depth."
> > >
> > > **Response:**
> > > We would like to clarify that the comparison requested by the reviewer is already included in the submission. In Fig. 3(c) and Fig. 4, we reapply the same residual block while holding the parameter budget fixed. The left panels of these subplots show that plain layer reuse has negligible effect on perplexity, while CEM’s within-layer recursion consistently improves performance. This demonstrates that the benefits of CEM are not attributable to increased depth alone.
> > >
> > > > "Test perplexity at matched FLOPs? and Training throughput and memory usage?"
> > >
> > > **Response:**
> > > Our model does not reduce FLOPs per forward pass. Because weight sharing primarily decreases parameter count, the main potential efficiency benefits of CEM relate to inference bandwidth usage rather than raw compute.
> > >
> > > Modern GPUs are typically memory–bound during decoding stage. The weight sharing in CEM, specifically in the key/value projections and the up/down MLP projections, allows the same weight tiles to be reused from on-chip SRAM, which may reduce memory movement and lower bandwidth pressure. While this suggests the possibility of improved inference throughput, realizing such gains would require specialized kernel implementations.
> > >
> > > We did not include throughput or memory-usage benchmarks in this paper, as these hardware-level evaluations and kernels are outside our current scope. We emphasize that any speedups at the hardware level are speculative at this stage. Nonetheless, CEM’s parameter efficiency provides a plausible path toward bandwidth-efficient implementations, which we view as a promising direction for future work.

---

> ### Comment · Reviewer_jswD · 2025-11-27
>
> Thanks for the reply! I've read them and I think those major issues are solved.
>
> I will increase my score to help the acceptance of the paper. Good luck!

---

> > ### Author Response · Authors · 2025-11-27
> >
> > Many thanks, Reviewer jswD!

---

### Official Review · Reviewer_j463 · 2025-11-01

**Soundness:** 3
**Presentation:** 3
**Contribution:** 4
**Rating:** 4
**Confidence:** 4

**Summary:**

This study utilizes a connection between Transformer and energy-based models. In particular, multi-head attention and gated MLP are two such components that can be viewed through the lens of energy-based models. This type of modeling yields a principled framework that formulates Transformer layer as solving an energy minimization problem, which also promotes interpretability. The proposed CEM is different from previous approaches in a sense that it takes account of input sequence history (i.e. causality). And weights are tied between projection operations, which leads to parameter efficient design.

**Strengths:**

Providing a new framework to view Transformer in more principled way. This can help interpretability and further optimization for popular Transformer architectures.

**Weaknesses:**

1. Reframing Transformer architecture with energy models will bring benefit such as weight sharing and thus parameter efficiency. However, this also takes away the freedom of standard Transformer architecture, where one can independently learn different weights.

2. Eq. in 116, if we look at the equation closely, it appears that in Fig. 1 top left, W_O and W_V should be switched.

3. 119, 'concatenated' and 116 eq. summation operator doesn’t match.

4. 180, The integration formulation here doesn't look correct. I believe one should define the integration from -\infty. (cf. cumulative distributions function and probability density function formulation) Also, GELU and SiLU both are non-zero for their negative domain. So I doubt this kind of construction is mathematically sound.

5. Eq. 7, Why should this thing be called energy term? This looks far-fetched to be called the energy term.

6. 188 paragraph, Authors empirically found that adding the diag term is more effective. However, A_k is learnable anyway. And I don't get the idea of combining both low-rankness and diagonality. If low-rankness doesn't help much and adding the diag term actually improved the performance, then it means imposing low-rankness from the first place is wrong.

7. 203, Similarly to #6, where's justification? Why UV and VU added together and why multiplying this preconditioned to the existing gradient enhances gradient descent? Plus, if enforcing semipositive definite is important, why not starting from Eigenvalue decomposition form with a diagnoal matrix in the middle (three matrices)?

8. 229-231, Future Work, Is it fair to not to include those looped Transformers in the experiments, as they are similar to CEM? Those references are published in 2019, 2023, and 2024 and authors should' taken them into account as well.

9. Fig. 1 bottom right, There's a recursion for CEM MLP. However, there's no related explanation in the text.

10. Alg. 1, MHA subroutine, last line of the inner for loop is problematic since it’s already in the for loop of k. In addition, the minus sign should be changed to the plus. See Sect. 2.1.

11. Alg. 1, Main Blk has RMSNorm inside the 2nd for loop. This is unnecessary since it is already inside the MLP subroutine. Also in the MLP subroutine, having RMSNorm inside the for loop is wrong, since it will normalize every iteration with a new t.

12. Similarly to #10, Alg. 1, MLP subroutine, the minus sign for the gradient descent step is wrong. See Sect. 2.2.


[editorial comments]
1. 461, recursion,can -> recursion, can
2. 473, yield -> yields
3. 'weights sharing' and 'weight sharing' are interchangeably used throughout the manuscipt. => be consistent
4. parameterize/paramerise => be consistent
5. Fig 1, weights such as W_Q => subscript to superscript change (i.e. W^Q) needed for consistency with later equations
6. 129, needs a curly brace pair for the set notation for A_k
7. 129, D_h feature dimension should've been appeared earlier after 120, along with a definition for D_r
8. Energy notation E mismatch throughout the manuscript: 60, Eq. 1, including \eta^\epsilon in Eq. 3
9. 719, Isn't -(1 / \tau) supposed to be -\tau? Cf. Eq. 1
10. 177, equations equation 6 and equation 10 -> equations 6 and 10
11. Eq. 12, RHS inside softmax, need the set notation cf. Eq. 4
12. Eq. 12, triangle should change to nabla, unless you want to define a new thing with \triangle
13. Eq. 12, Superscript \epsilon actually should be the name of the function
14. Same problem for Eq. 13
15. 215, could be trained provide -> could be trained to provide
16. 222, 227, and Alg. 1 have \eta's with subscripts to denote different energy function name, however Eqs. 3 and 9 have \eta's with superscripts.
17. Alg. 1, MHA subroutine, Delete the superscript k from the 1st line. This is because in the main block, MHA is only invoked after all K iterations.
18. 269, netowrks -> networks

**Questions:**

Please see my major comments above. I could've given a higher score if I hadn't found that many issues.

---

> ### Author Response · Authors · 2025-11-20
> **Author Response to Reviewer j463 (Part 1)**
>
> Thank you very much for your thoughtful and detailed review. We greatly appreciate the time and effort you invested in evaluating our work. Below we provide detailed responses to your major comments, and all other editorial issues have already been addressed in the revised version uploaded.
>
> >"Reframing Transformer architecture with energy models will bring benefit such as weight sharing and thus parameter efficiency. However, this also takes away the freedom of standard Transformer architecture, where one can independently learn different weights."
>
> **Response:**
> We agree that weight sharing imposes parameterisation constraints compared to a standard Transformer, where each projection matrix is independently learned. We highlight several points to clarify why this does not reduce model expressivity in practice and can even be a favorable parameterization:
>
> **(1) Weight sharing enables scaling to larger effective model sizes under a fixed parameter budget**
> Sharing parameters reduces memory footprint and communication overhead. This allows us to allocate the saved parameters to increase hidden dimension, number of heads, etc. All of which expand expressiveness. In other words, CEM models of the same parameter budget can afford a wider or deeper representation space, and thus remains as expressive.
>
> **(2) Weight sharing improves hardware efficiency by reducing high-cost memory traffic**
> Modern GPU accelerators are sometimes bottlenecked not by FLOPs but by SRAM-HBM data movement, especially during the decoding stage. A CEM block shares key/value projection weights, and up/down projection weights, meaning: The same weight tiles are reused from fast on-chip SRAM for both key/values and up/down projections. This results in significantly lower memory bandwidth requirements so that one can load larger tiles into SRAM each time and dramatically speed up computation.
> We did not explicitly discuss this in the manuscript because to show this advantage, one needs to write customized kernels, which is beyond the scope of this paper. But we see this as a practical direction to deliver real impact using our CEM framework.
>
> > "Eq. in 116, if we look at the equation closely, it appears that in Fig. 1 top left, W_O and W_V should be switched."
>
> > "119, 'concatenated' and 116 eq. summation operator doesn’t match."
>
> We have carefully re-examined the manuscript and believe that both instances in the original paper are correct. Let us elaborate in more detail.
> In standard multi‑head attention, the outputs from each head are concatenated,
> $$O = [O_1^\top, \dots, O_K^\top]^\top
> $$ and then passed through the output projection. This can be expressed as
> $$W^{O\top} O
> = [W^{O\top}_1, \dots, W^{O\top}_K] [O_1^\top, \dots, O_K^\top]^\top = \sum_k W^{O\top}_k O_k
> $$ Thus, the concatenation form and the summation form are mathematically equivalent, and we adopt the latter for a cleaner presentation. We have added a detailed explanation in Appendix A1 in case this isn't clear.
> Regarding Fig. 1, the ordering of $W^O$ and $W^V$ is intentional. The gray bar denotes the hidden state $h$. First, $W^V$ maps $h$ to the value vector $v$, which is then projected back using $W^O$. The figure illustrates the computation for a single attention head. It may also be helpful to note that the Softmax operation produces a scalar weight, while the transformations with $W^O$ and $W^V$ yield feature vectors of the same dimension, scaled by the Softmax weight. We have updated the figure caption to make this clearer.
>
> > "180, The integration formulation here doesn't look correct. I believe one should define the integration from -\infty. (cf. cumulative distributions function and probability density function formulation) Also, GELU and SiLU both are non-zero for their negative domain. So I doubt this kind of construction is mathematically sound."
>
> **Response:**
> We appreciate the reviewer’s careful observation. The construction is indeed mathematicaly correct, though we agree that defining the integration from $-\infty$ would make the formulation clearer and avoid potential confusion. We have updated this in the revised text.
> Note that our requirement is simply that $\phi'(x)$ corresponds to $\text{SiLU}(x)$ or $\text{GeLU}(x)$. That is,
> $$
> \phi'(x) = \left(\int_{-\infty}^{x} \sigma(z)\,dz\right)' = \sigma(x).
> $$ Even if one writes
> $$
> \phi(x) = \int_{0}^{x} \sigma(z)\,dz = \int_{-\infty}^{x} \sigma(z)\,dz - \int_{-\infty}^{0} \sigma(z)\,dz,
> $$ the derivative with respect to $x$ still yields $\sigma(x)$. Thus the construction is sound, though in the revised manuscript we have already adopted the $-\infty$ lower limit to improve clarity.

---

> > ### Comment · Reviewer_j463 · 2025-11-22
> > **Thank you for the response**
> >
> > I've just finished reviewing Part 1 and I note that my concerns up to #4 were addressed.

---

> ### Author Response · Authors · 2025-11-20
> **Author Response to Reviewer j463 (Part 2)**
>
> > "Eq. 7, Why should this thing be called energy term? This looks far-fetched to be called the energy term."
>
> **Response:**
> We respectfully disagree - this formulation looks a lot like the associative memory form of Hopfield Networks. We can re-write eq. (7) as
>
> $$
> E(x|h_i) = \sum_{j=1}^{D_m} -\alpha_{ij}\sigma(v_j^\top x),
> $$
>
> This matches the form of Eq. 9 in *[Hopfield Networks is All You Need](https://ml-jku.github.io/hopfield-layers/)*, where we recognise v_j as an associative memory.
>
> Under this perspective, the gates $\alpha_{ij} = w_j^\top h_i$ act as input-dependent coefficients on each pattern vector.
>
> While one could define other energy terms that more closely resemble traditional energy functions in physics, our focus in this paper is to start from existing transformer architectures. For this reason, we do not explore alternative energy formulations that may be more naturally expressed as energy.
>
> > "Authors empirically found that adding the diag term is more effective. However, $A_k$ is learnable anyway. And I don't get the idea of combining both low-rankness and diagonality."
>
> **Response:**
> The diagonal-plus-low-rank (D+LR) parameterization is a widely used approximation that provides better expressive power over purely low-rank matrices, while keeping computational cost and memory low. It provides a strong approximation to a full matrix at $O(D + Dr)$ cost instead of $O(D^2)$. It is therefore strictly more expressive than either component independently, yet far cheaper than a full $D \times D$ matrix. It is not equivalent to learning a full matrix. Diagonal-plus-low-rank (D+LR) has been used in for example [Annotated S4](https://iclr-blog-track.github.io/2022/03/25/annotated-s4/#step-3-diagonal-plus-low-rank) or [Low-rank plus diagonal approximations for Riccati-like matrix
>  differential equations](https://arxiv.org/abs/2407.03373).
>
> > "Why UV and VU added together and why multiplying this preconditioned to the existing gradient enhances gradient descent? Plus, if enforcing semipositive definite is important, why not starting from Eigenvalue decomposition form with a diagonal matrix in the middle (three matrices)?"
>
> **Response:**
> The use of preconditioners originates from the optimization literature, particularly in second‑order methods. Newton’s method can dramatically reduce the number of steps to reach a well‑optimized state compared to pure gradient descent, and it has the following update form:
> $$
> x_{t+1} = x_t - H^{-1} \nabla f(x_t),
> $$ where $H$ is the Hessian matrix of the objective function. Since computing and inverting the Hessian is generally expensive, a wide variety of approximations have been proposed. These approximations are referred to as *preconditioner matrices*, which are designed to capture the curvature information and speed up optimization. One important constraint of a preconditioner matrix is that it has to be positive semidefinite (p.s.d.). p.s.d. preconditioners ensure that the update direction decrease the energy for sufficiently small step sizes.
> In our paper, the term $P = \text{diag}(d) + UV + VU$ is p.s.d. by construction.
>
> Regarding the reviewer’s suggestion to start from an eigenvalue-decomposition form, we note that although the decomposition
> $$
> M = Q \Lambda Q^\top
> $$ with diagonal $\Lambda$ and orthogonal $Q$ is theoretically valid, computing $Mx$ requires $\mathcal{O}(n^2)$ operations. In contrast, our low-rank construction enables computing $Px$ in $\mathcal{O}(nr)$, where $r$ is the rank of $U$ and $V$.
>
> Our parameterization also offers practical advantages for initialization: setting $V = 0$ and $d = 1$ yields $P = I$, the identity matrix, providing a stable starting point. In the eigen-decomposition parameterization, by contrast, one would need to enforce constraints such as orthogonality of $Q$ and positivity of $\Lambda$ during training, which complicates optimization.
>
> (edit: fixed typo in equation)

---

> > ### Author Response · Authors · 2025-11-20
> > **Author Response to Reviewer j463 (Part 3)**
> >
> > > "Is it fair to not include those looped Transformers in the experiments, as they are similar to CEM?"
> >
> > **Response:**
> > Note that although looped transformers and CEM blocks both involve recursions, they operate in fundamentally different ways.
> > Looped Transformers repeatedly apply deep Transformer blocks (MHA → MLP → MHA → MLP ...), usually consists of many such layers. CEM recursion, by contrast, performs *within‑layer* recursion by unrolling the gradient‑based update inside a single MHA or MLP block. Importantly, CEM recursion is also not the same as reusing a single layer: it updates only the optimized state $x$ while keeping the input $h$ fixed. As a result, the gates and key/value projections are computed once and then reused across recursive steps.
> > In short, looped-layer recursion and CEM’s within-layer recursion operate along different "dimensions". They are orthogonal rather than substitutes. Studying how these two forms of recursion might interact or be combined is an interesting direction for future work.
> >
> > In addition, we provided experimental results in Fig. 3(c) and Fig. 4 showing that CEM recursion is more effective than plain layer reuse (left panel of each subfigure), where the model simply applies the same layer twice.
> >
> > > "Fig. 1 bottom right, There's a recursion for CEM MLP. However, there's no related explanation in the text."
> >
> > **Response:**
> > The recursion is explained in Section 2.3 (Multiple Recursion Steps), and we have now added an explicit reference to this section in the figure caption.
> >
> > > Weaknesses 10-12
> >
> > > All the other editorial comments 1-18
> >
> > **Response:**
> > Thank you very much for these detailed editorial suggestions. We have incorporated all of them into the revised version, with a few exceptions that we would like to clarify:
> >
> > 1. **Regarding weakness 11:** The use of RMSNorm is intentional. The RMSNorm in the main block is applied to normalize the input feature $h$, while the RMSNorm inside the MLP subroutine normalizes the free variable $x$ being optimized. This ensures that the optimization dynamics evolve on a hypersphere, which is important for stability.
> >
> > 2. **Regarding editorial comment 12:** The use of the triangle symbol $\Delta$ is also intentional. In our notation, the triangle $\Delta$ denotes the *preconditioned update*, which is distinct from the gradient operator $\nabla$. We have clarified this distinction in the manuscript to avoid confusion.

---

> > > ### Comment · Reviewer_j463 · 2025-11-24
> > > **Ack for Part 3**
> > >
> > > Finished reviewing Part 3 and I appreciate authors' effort for addressing all my remaining concerns.
> > >
> > > Quick suggestion for Eqs. 12 and 13 (nabla and triangle thing): I suggest authors use "define" notation such as :=, instead of =, to clearly define the new definitions.
> > >
> > > I am willing to raise my score and will decide how much after I hear back from the authors for Part 2 questions.

---

> > > > ### Author Response · Authors · 2025-11-25
> > > > **Further Response to Reviewer j463**
> > > >
> > > > Thank you for the helpful suggestions.
> > > >
> > > > Following your feedback, we have made the following updates:
> > > >
> > > > 1. Added a new appendix section (A.2) providing background on the diagonal-plus-low-rank parameterisation.
> > > > 2. Replaced “=” with “:=” when introducing the new triangular notation.
> > > > 3. Ran ablation experiments on the CEM model (dimension-matched to the 162M Llama model) using only the diagonal component and only the low-rank component, and compared them to the full diagonal-plus-low-rank parameterisation. The results show a substantial performance drop when using only the diagonal component and a smaller drop when using only the low-rank component, while the full diagonal-plus-low-rank form performs the best.
> > > >
> > > > | Model     | Perplexity (Lower is better)  |
> > > > | --------- | ----------- |
> > > > | Diag Only | 27.89702354 |
> > > > | LR Only   | 16.03991014 |
> > > > | Diag + LR | 15.61496891 |
> > > >
> > > > We hope the revision and the additional results address the reviewer’s remaining concern.

---

> > ### Comment · Reviewer_j463 · 2025-11-22
> > **Ack for Part 2**
> >
> > Just finished reviewing Part 2 and I appreciate the detailed response.
> >
> > As for #6, I'm still curious if diagonal matrix only would be better or D+LR structure would bring more advantage to CEM. Plus, I suggest authors add some justification in the manuscript that justifies the usage of D+LR structure along with said references.

---

### Author Response · Authors · 2025-12-02
**Summary of Rebuttal Discussion**

We sincerely thank all reviewers and ACs for their time and efforts in evaluating and improving our work. We are grateful for the careful assessments and constructive feedback provided throughout the rebuttal/discussion period. This comment serves as our final remark and summarizes the status of the submission following the discussion.

During the discussion, reviewers j463 and jswD explicitly confirmed that their major concerns had been addressed and indicated that they had raised (or agreed to raise) their scores. Below is a summary of the key points from the rebuttal:

**Technical questions**

To address the technical questions raised by the reviewers, we provide detailed explanation and derivation in the rebuttal and have also expanded Appendix A with additional background and details. In particular, we explained the equivalence between concatenation and summation forms in multi-head attention; showed how our energy functions connect GELU/SiLU; justification for why these constructions constitute energy terms; clarifications on the role/constraints/parameterization of preconditioners; how within-layer recursions are different from looped transformers; and so on. Reviewers j463 and jswD agreed that these answered their questions about the technical details.

**Editorial issues**

All editorial issues raised by reviewers j463 and 5RHo have been addressed point-by-point in the revised version. Where comments stemmed from misunderstandings (e.g., attention summation vs. concatenation, notation for energies and updates), we clarified the underlying technical details in the rebuttal and made it clearer in the paper.

**Ablation studies**

We expanded our ablations to directly address the reviewers’ concerns.

* In the reponse to reviewer j463, we provide new experiments comparing pure diagonal, pure low-rank, and full diagonal-plus-low-rank (D+LR) parameterizations; both “diag only” and “low-rank only” significantly degrade performance, while full D+LR performs best, supporting our chosen design.

* We highlight that existing results (Fig. 3(c) and Fig. 4) already compare CEM recursion (T > 1) against naive layer reuse under a matched parameter budget: simply reapplying the same residual block yields negligible gains, whereas CEM’s within-layer recursion consistently improves perplexity, demonstrating that the benefits are not attributable to increased depth alone.

* We added Gaussian-process regression experiments to study CEM-MLP recursion for T = 1, 2, 4, 8 in a controlled setting. Increasing T consistently improves performance with larger T. While we acknowledge that we have not yet found a fully robust recipe for stable recursion at large T in transformers, we consistently observe improvements for T = 2. See Appendix Table 3.

**Larger-scale experiments**

We also extended our empirical study beyond the original ~80–160M range. New 256M-scale experiments (see Appendix Fig 5) show that a 205M-parameter CEM model can outperform a 256M Llama baseline, providing evidence that our approach continue to scale beyond the smallest models. At the same time, we are transparent that full 7B+ pretraining is beyond our current computational budget, and we position this work as introducing and validating the CEM framework at mid scale, with larger-scale extensions left to future work.

**Contribution, energy perspective, and practical benefits**

Reviewers agreed that CEM provides a principled and unified energy-based view of both multi-head attention and gated MLPs, extending beyond prior Hopfield-style interpretations that focus only on attention. In the rebuttal, we clarified that the energy formulation is used as a lens for layer design, grounded in associative-memory and retrieval perspectives, rather than replacing the original training objectives.

Several architectural extensions follow naturally and are validated empirically: multi-step updates improve expressiveness under a fixed parameter budget; per-head preconditioners offer lightweight curvature information; and diagonal distance matrices capture token interactions more flexibly. Each contributes consistent gains in our language-modeling experiments.

The framework also induces parameter-efficient weight sharing in both MHA and MLP blocks, such as shared key/value and up/down projections. This reduces the static memory footprint and encourages on-chip reuse of weights, which, with appropriate kernel support, can potentially translate into meaningful bandwidth and efficiency benefits.

We hope these discussions have helped clarify the key points and that the revised manuscript now presents a clearer and more polished version of our work.

---

### Note · Program_Chairs · 2026-01-17
**Submission Desk Rejected by Program Chairs**

The following references in this submission do not refer to real documents and/or have major errors in bibliographic information:

 Shipeng Zhai and et al. Multi-head latent attention for efficient transformers. In NeurIPS, 2023. URL https://arxiv.org/abs/2305.09828